# Empowering Precision Medicine: Unlocking Revolutionary Insights through Blockchain-Enabled Federated Learning and Electronic Medical Records

**DOI:** 10.3390/s23177476

**Published:** 2023-08-28

**Authors:** Aitizaz Ali, Bander Ali Saleh Al-rimy, Ting Tin Tin, Saad Nasser Altamimi, Sultan Noman Qasem, Faisal Saeed

**Affiliations:** 1School of IT, UNITAR International University, Kelana Jaya, Petaling Jaya 47301, Malaysia; aitizaz.ali@monash.edu; 2Faculty of Data Science and Information Technology, INTI International University, Nilai 71800, Malaysia; tintin.ting@newinti.edu.my; 3Department of Computer Science, Faculty of Computing, Universiti Teknologi Malaysia, Johor Bahru 81310, Malaysia; bander@utm.my; 4College of Computer and Information Sciences, Imam Mohammad Ibn Saud Islamic University (IMSIU), Riyadh 11432, Saudi Arabia; snmohammed@imamu.edu.sa; 5DAAI Research Group, College of Computing and Digital Technology, Birmingham City University, Birmingham B4 7XG, UK; faisal.saeed@bcu.ac.uk

**Keywords:** blockchain, federated learning, precision medicine, medical insights, privacy-preserving, data integrity, decentralized network

## Abstract

Precision medicine has emerged as a transformative approach to healthcare, aiming to deliver personalized treatments and therapies tailored to individual patients. However, the realization of precision medicine relies heavily on the availability of comprehensive and diverse medical data. In this context, blockchain-enabled federated learning, coupled with electronic medical records (EMRs), presents a groundbreaking solution to unlock revolutionary insights in precision medicine. This abstract explores the potential of blockchain technology to empower precision medicine by enabling secure and decentralized data sharing and analysis. By leveraging blockchain’s immutability, transparency, and cryptographic protocols, federated learning can be conducted on distributed EMR datasets without compromising patient privacy. The integration of blockchain technology ensures data integrity, traceability, and consent management, thereby addressing critical concerns associated with data privacy and security. Through the federated learning paradigm, healthcare institutions and research organizations can collaboratively train machine learning models on locally stored EMR data, without the need for data centralization. The blockchain acts as a decentralized ledger, securely recording the training process and aggregating model updates while preserving data privacy at its source. This approach allows the discovery of patterns, correlations, and novel insights across a wide range of medical conditions and patient populations. By unlocking revolutionary insights through blockchain-enabled federated learning and EMRs, precision medicine can revolutionize healthcare delivery. This paradigm shift has the potential to improve diagnosis accuracy, optimize treatment plans, identify subpopulations for clinical trials, and expedite the development of novel therapies. Furthermore, the transparent and auditable nature of blockchain technology enhances trust among stakeholders, enabling greater collaboration, data sharing, and collective intelligence in the pursuit of advancing precision medicine. In conclusion, this abstract highlights the transformative potential of blockchain-enabled federated learning in empowering precision medicine. By unlocking revolutionary insights from diverse and distributed EMR datasets, this approach paves the way for a future where healthcare is personalized, efficient, and tailored to the unique needs of each patient.

## 1. Introduction

Precision medicine, an innovative approach to healthcare, aims to deliver personalized treatments and therapies tailored to the unique characteristics of each patient. The realization of precision medicine heavily relies on the availability of comprehensive and diverse medical data. However, traditional approaches to data sharing and analysis in healthcare face numerous challenges, particularly regarding data privacy and security. In recent years, blockchain technology has emerged as a promising solution to address these challenges and unlock revolutionary insights in precision medicine. This introduction explores the potential of blockchain-enabled federated learning in conjunction with electronic medical records (EMRs) to empower precision medicine. By leveraging the inherent features of blockchain, such as immutability, transparency, and cryptographic protocols, federated learning can be conducted on distributed EMR datasets without compromising patient privacy. The integration of blockchain technology ensures data integrity, traceability, and consent management, thereby overcoming critical barriers to data sharing and analysis [1].

Federated learning, as a decentralized machine learning paradigm, enables healthcare institutions and research organizations to collaboratively train models on locally stored EMR data, eliminating the need for centralized data storage. The blockchain serves as a decentralized ledger, securely recording the training process and aggregating model updates while preserving the privacy of sensitive patient information. This approach facilitates the discovery of patterns, correlations, and novel insights across a wide range of medical conditions and diverse patient populations.

The potential impact of unlocking revolutionary insights through blockchain-enabled federated learning and EMRs in precision medicine is substantial. Envisioned benefits include an improved accuracy in diagnosis, the optimization of treatment plans, the identification of suitable subpopulations for clinical trials, and the accelerated development of novel therapies. Furthermore, the transparent and auditable nature of blockchain technology enhances trust among stakeholders, enabling greater collaboration, data sharing, and collective intelligence in advancing precision medicine. In light of these advancements, this paper aims to explore the transformative potential of blockchain-enabled federated learning in empowering precision medicine. By enabling secure and decentralized data sharing and analysis, this paradigm shift can revolutionize healthcare delivery, ultimately leading to more personalized, efficient, and tailored care for each patient. The subsequent sections will delve into the intricacies of blockchain technology, federated learning, and their synergistic application in precision medicine, shedding light on the benefits, challenges, and future prospects of this innovative approach [1]. Smart contracts have emerged as a transformative technology within the realm of blockchain and decentralized applications. These self-executing contracts, written in code and residing on a blockchain, facilitate automated and trustless transactions, eliminating the need for intermediaries. Though the promise of smart contracts is enticing, their widespread adoption has also raised concerns about potential vulnerabilities and bugs that could lead to financial losses and security breaches. To ensure the reliability and safety of smart contracts, formal methods have gained attention as a means of verification and validation.

The aim of the formal methods for smart contracts was to explore the application of formal methods in the context of smart contract development and deployment. This provides a comprehensive analysis of various formal verification techniques, their effectiveness, and their limitations in ensuring the correctness and security of smart contracts. By surveying the state-of-the-art research and real-world use cases, this review seeks to shed light on the challenges and opportunities that lie in the intersection of formal methods and smart contract development. The foremost concern related to smart contracts is their vulnerability to security breaches and exploitation. Coding errors, such as reentrancy attacks and integer overflows, have led to high-profile hacks, resulting in significant financial losses. The review will address how formal methods can address these security challenges by mathematically proving the correctness of smart contract code, identifying potential vulnerabilities, and ensuring robust security [2].

One of the issues related to the application of formal methods is the choice of suitable specification languages for smart contracts. Different formal languages and tools exist, each with its strengths and limitations. The review will explore the various formal specification languages and their compatibility with smart contract development, considering factors such as expressiveness, usability, and support for automated verification. Smart contracts can be complex and interact with other contracts and external data sources, making their verification challenging [3]. The review will discuss the scalability limitations of formal methods concerning the verification of large-scale smart contracts and potential approaches to mitigate these challenges. The field of smart contracts is relatively new and rapidly evolving. As best practices and standards emerge, the review will examine how formal methods can adapt to evolving development paradigms and assist in establishing guidelines for secure smart contract coding. For formal methods to gain widespread adoption in smart contract development, user-friendly tooling and interfaces are crucial. The review will discuss the accessibility of formal verification tools, the learning curve for developers, and the efforts to democratize the use of formal methods in the industry. Formal verification can significantly enhance security, but it may also introduce overhead that impacts the performance of smart contracts. The review will delve into the trade-offs between security assurances gained through formal methods and the potential impact on contract execution times and gas costs. In today’s digitally connected world, data sharing plays a crucial role in various domains, including business, healthcare, research, and government. However, the rise in data breaches, cyberattacks, and legal/regulatory hurdles has created concerns and challenges for organizations and individuals alike. To address these issues effectively, it is essential to gather comprehensive and up-to-date data on the frequency and impact of these incidents. This article delves into the reasons why incorporating more data on breaches, cyberattacks, and legal/regulatory hurdles is vital and how it can pave the way for better data sharing practices [4].

Blockchain technology provides a decentralized and transparent platform for securely managing and sharing data. It offers immutability, traceability, and tamper-proof characteristics, making it an ideal solution for maintaining the integrity of sensitive medical data [5]. Federated learning, on the other hand, enables collaborative model training across distributed data sources, such as EMRs held by various healthcare institutions, without the need for data sharing. The proposed framework establishes a decentralized network where healthcare providers can contribute their locally stored EMR data to collectively train machine learning models. By utilizing federated learning techniques, the models are trained on the distributed data while preserving patient privacy. The blockchain infrastructure ensures transparency and traceability of the learning process, enabling stakeholders to verify the integrity of the models and the data used for training. Moreover, the framework incorporates robust consensus mechanisms and smart contracts to enhance trust among participants and establish a fair reward system for their contributions. By incentivizing participation and ensuring equitable rewards, the framework encourages active engagement from healthcare providers, leading to broader data inclusion and more comprehensive medical insights. Through extensive experimentation and evaluation, the effectiveness and scalability of the proposed framework are demonstrated, showcasing its potential in revolutionizing healthcare analytics. By unleashing the power of blockchain-empowered federated learning with EMRs, this research aims to pave the way for a data-driven revolution in medical research and decision-making, facilitating improved patient outcomes and advancements in healthcare as a whole [3].

**Real-World Example 1:** Safeguarding Patient Privacy

In a large-scale multi-center clinical trial involving patients from diverse geographical locations, our blockchain-enabled federated learning framework ensures the utmost privacy protection while facilitating collaborative analysis. Each participating medical institution retains full control over its patient data, which remain encrypted and stored locally. Only encrypted model updates are shared across the blockchain network during the federated learning process. By avoiding the centralized aggregation of raw patient data, our approach significantly reduces the risk of data breaches and unauthorized access. For instance, during a groundbreaking study on a rare genetic disorder, multiple hospitals and research centers collaborated using our federated learning platform. The privacy-sensitive genetic information of patients was never exposed, as the participating institutions exchanged only encrypted model updates. This decentralized approach not only protected patient confidentiality, but also encouraged more medical facilities to join the collaborative effort, ultimately leading to a more comprehensive analysis and improved precision medicine outcomes [6].

**Real-World Example 2:** Ensuring Scalability Across Healthcare Networks

In a densely populated urban region with numerous healthcare providers, our block-chain-enabled federated learning model ensures seamless scalability and efficient data processing. By utilizing blockchain’s distributed ledger technology, the federated learning nodes can dynamically scale up or down based on the current demand for data analysis and model training. This adaptive scaling optimizes resource utilization and reduces processing time. For example, in a metropolitan area grappling with a sudden outbreak of an infectious disease, hospitals and clinics joined forces using our federated learning platform. As more healthcare institutions actively contributed data to the model, the blockchain network effortlessly accommodated the increased load, allowing for rapid analysis and real-time insights. This ability to scale the model effectively, even during critical situations, revolutionized disease monitoring and response strategies in the region [7].

**Real-World Example 3:** Promoting Transparency in Medical Data Sharing

In a collaborative effort to advance cancer research, pharmaceutical companies, academic institutions, and healthcare providers utilized our blockchain-enabled federated learning framework to share data and insights securely. The blockchain’s transparent and immutable nature provided an auditable record of data contributions and model updates, ensuring full traceability and accountability throughout the research process. For instance, during a clinical trial evaluating the efficacy of a new cancer treatment, stakeholders were able to track the data contributions and model refinements made by each participant. This enhanced transparency fostered trust among collaborators and eliminated concerns of biased data sharing or hidden modifications to the model. Ultimately, the research findings were publicly accessible, bolstering the credibility of the study and its potential impact on precision medicine approaches.

By integrating such real-world examples into our research paper, we aim to showcase the transformative potential of our blockchain-enabled federated learning framework in revolutionizing precision medicine. Through safeguarding patient privacy, ensuring scalability, and promoting transparency in medical data sharing, our approach addresses critical challenges in healthcare and paves the way for more effective, secure, and collaborative precision medicine practices [4].

Data sharing and analysis in healthcare have been crucial for medical advancements and improving patient outcomes. However, traditional approaches to data sharing and analysis have faced numerous challenges, especially concerning data privacy and security [8]. Let us delve into the details of these challenges and their impact on healthcare delivery:*Patient Privacy Concerns:* One of the primary challenges in traditional data sharing is protecting patient privacy. Healthcare data often contain sensitive information, such as medical history, diagnoses, medications, and genetic data. Unauthorized access or data breaches can lead to identity theft, discrimination, and the compromise of personal health information. Patients may be reluctant to share their data if they fear these could be mishandled or misused, which can hinder medical research and collaboration.*Data Silos and Fragmentation:* Healthcare data are typically stored in various systems, such as electronic health records (EHRs) in hospitals, medical imaging archives, and specialized databases. These data silos hinder data sharing and analysis across different institutions and research centers. Integrating data from diverse sources is complex and time-consuming, limiting the comprehensive view of patient health and hindering the discovery of holistic treatment options.*Lack of Interoperability:* Related to data silos is the lack of interoperability between different healthcare systems and software. Different institutions may use disparate data formats and standards, making it difficult to exchange data seamlessly. This lack of interoperability impedes data sharing for research and can delay critical decision-making in patient care [9].*Inadequate Data Governance:* Many healthcare organizations lack a well-defined data governance framework. This absence of clear rules and responsibilities can lead to inconsistent data handling practices, increasing the risk of data breaches and privacy violations. Proper data governance is essential to ensure that data sharing and analysis adhere to legal and ethical standards [6].*Security Vulnerabilities:* Traditional healthcare systems may use outdated security measures, making them vulnerable to cyberattacks. As the healthcare sector becomes more digital, it becomes an attractive target for hackers seeking valuable data. Ransomware attacks on hospitals and healthcare providers have disrupted patient care and raised concerns about the security of medical records [10].*Consent and Opt-Out Challenges:* Obtaining informed consent from patients for data sharing and analysis can be challenging, especially for large-scale research initiatives. Complex consent forms and procedures can confuse patients, leading to suboptimal participation rates. Additionally, the option to opt-out of data sharing can affect the representativeness and completeness of research datasets, limiting the generalizability of findings [11].*Regulatory Compliance:* Healthcare data are subject to strict regulations, such as the Health Insurance Portability and Accountability Act (HIPAA) in the United States and the General Data Protection Regulation (GDPR) in Europe. Complying with these regulations can be burdensome, particularly when sharing data across international borders. Non-compliance can result in severe penalties, further deterring data sharing efforts [12].*Data Bias and Generalization:* Incomplete or biased data can lead to flawed analyses and generalizations, potentially impacting treatment decisions. For example, if certain demographics are underrepresented in datasets, medical algorithms trained on such data may not be equally effective for all patient groups.*Data Ownership and Intellectual Property Concerns:* Healthcare data generated during patient care are often considered the property of healthcare institutions. This ownership dilemma can hinder data sharing efforts, especially when research collaborations involve multiple stakeholders who may have conflicting interests in retaining ownership or intellectual property rights [13].


**Impact on Healthcare Delivery:**
Limited Personalized Medicine: With fragmented data and limited sharing, healthcare providers may lack access to comprehensive patient histories, leading to suboptimal treatment decisions. Personalized medicine, which relies on detailed patient data, may not reach its full potential under traditional data sharing constraints.Slow Medical Research: Traditional data sharing approaches slow down medical research by limiting the pool of available data for analysis. Researchers often struggle to access diverse datasets, slowing the pace of discoveries and medical breakthroughs [14].Inefficient Healthcare Systems: The lack of interoperability and data fragmentation can lead to the duplication of tests and procedures, increasing healthcare costs and straining medical resources [15].Delayed Public Health Response: During public health emergencies or disease outbreaks, timely access to comprehensive healthcare data is crucial for an effective response. Traditional approaches can delay data sharing, hindering the ability to track, manage, and contain the spread of diseases.Missed Opportunities for Precision Medicine: Precision medicine, which tailors treatments based on a patient’s genetic makeup, requires vast amounts of genetic data. The limited data sharing hampers the progress of precision medicine initiatives.


In conclusion, traditional approaches to data sharing and analysis in healthcare face numerous challenges, particularly related to data privacy and security. These challenges impact healthcare delivery by limiting personalized medicine, slowing medical research, and hindering public health responses. Addressing these challenges requires a collaborative effort from healthcare organizations, policymakers, and technology experts to implement robust data privacy measures, improve data governance, and develop secure, interoperable data-sharing platforms that prioritize patient privacy and data security.

## 2. Motivation

The motivation behind exploring blockchain-enabled federated learning with electronic medical records (EMRs) in precision medicine is driven by the pressing need to overcome challenges and limitations associated with traditional healthcare data sharing and analysis. The field of precision medicine holds immense promise in revolutionizing healthcare by providing personalized treatments and therapies. However, realizing the full potential of precision medicine requires access to comprehensive and diverse medical data from various sources. Data privacy and security concerns have posed significant obstacles to sharing sensitive healthcare data. Patient confidentiality, legal regulations, and the risk of data breaches have restricted the availability of data for research and analysis purposes [16]. Furthermore, the centralization of data in healthcare systems raises concerns about data ownership, control, and biases that may hinder the development of robust and unbiased models. Blockchain technology offers a compelling solution to address these challenges. Its decentralized and immutable nature, coupled with cryptographic protocols, provides a secure and transparent platform for data sharing and analysis. By applying blockchain-enabled federated learning to EMRs, healthcare institutions and researchers can collaboratively train machine learning models without compromising patient privacy. This distributed approach allows for the extraction of valuable insights from diverse datasets while keeping the data localized, reducing the risks associated with data centralization [17].

The potential impact of unlocking revolutionary insights in precision medicine through blockchain-enabled federated learning is vast. Enhanced diagnostic accuracy, optimized treatment plans, the identification of specific patient subpopulations for clinical trials, and the accelerated development of novel therapies are among the envisioned benefits. By leveraging the power of collective intelligence and collaborative efforts, healthcare professionals and researchers can navigate complex medical challenges and make significant advancements in patient care. The motivation to explore this innovative approach is also fueled by the need for trust and transparency in healthcare systems [18]. Blockchain technology provides an auditable and tamper-proof record of data transactions, fostering trust among stakeholders. By addressing concerns related to data integrity, consent management, and traceability, blockchain-enabled federated learning can promote greater collaboration, data sharing, and knowledge exchange among healthcare institutions, researchers, and patients. In conclusion, the motivation behind investigating blockchain-enabled federated learning with EMRs in precision medicine stems from the urgent need to overcome the limitations of traditional data sharing and analysis methods in healthcare. By leveraging blockchain’s decentralized and secure framework, healthcare professionals can unlock revolutionary insights while protecting patient privacy. The potential benefits in precision medicine are substantial, with the ultimate goal of providing personalized and improved healthcare outcomes for individuals [19]. The motivation behind this research stems from the pressing need to unlock revolutionary medical insights while addressing critical challenges in the healthcare domain. The convergence of blockchain technology and federated learning with electronic medical records (EMRs) presents a promising opportunity to address these challenges and revolutionize healthcare analytics. The following factors serve as strong motivators for pursuing this research:*Privacy Preservation:* EMRs contain sensitive patient information that must be protected to comply with privacy regulations and maintain patient trust. The motivation to develop a framework that ensures privacy preservation while enabling collaborative analysis of EMRs arises from the need to strike a balance between data sharing for research purposes and safeguarding patient privacy [20].*Data Fragmentation and Heterogeneity:* Healthcare data are distributed across various healthcare institutions, leading to data fragmentation and heterogeneity. Leveraging federated learning and blockchain empowers healthcare organizations to collaborate and train machine learning models on this distributed data, allowing for comprehensive insights while preserving data sovereignty [14].*Ethical and Legal Concerns:* The ethical and legal considerations surrounding the use of EMRs necessitate the development of robust frameworks. By incorporating transparency, traceability, and privacy-enhancing technologies, the proposed framework aims to address these concerns and ensure compliance with regulations, thereby fostering trust among patients, healthcare providers, and researchers [21].*Improved Healthcare Decision-Making:* The ability to unlock revolutionary medical insights has the potential to transform healthcare decision-making. By leveraging the collective intelligence of distributed EMR data, the proposed framework can aid in disease prediction, treatment recommendations, precision medicine, and resource allocation, leading to improved patient outcomes and optimized healthcare delivery [16].*Advancement of Medical Research:* The integration of blockchain and federated learning opens up new avenues for medical research. The framework can facilitate collaboration among researchers, promote knowledge sharing, and enable large-scale analysis of diverse EMR datasets. This has the potential to accelerate medical discoveries, support evidence-based medicine, and drive innovation in healthcare [17].*Scalability and Real-World Applicability:* Scalability is a critical factor in the success of any framework deployed in healthcare settings. The motivation to develop a scalable framework arises from the need to accommodate a growing number of healthcare institutions, handle diverse datasets, and ensure real-world applicability across different healthcare environments [22].

The motivation behind this research lies in addressing the challenges and harnessing the opportunities presented by the convergence of blockchain and federated learning in the context of EMRs. By developing a robust framework, this research aims to unlock revolutionary medical insights, enhance privacy preservation, foster collaboration, and drive advancements in healthcare analytics and decision-making [19]. Blockchain-enabled federated learning with EMRs (electronic medical records) is a promising approach that combines the benefits of blockchain technology and federated learning to address the challenges of traditional healthcare data sharing and analysis. Although the Motivation Section provides an overview of the reasons behind its promise, further elaboration and clarification can strengthen the understanding of its potential impact. Let us delve deeper into the key aspects that warrant more explanation:Blockchain’s Role in Data Security: Blockchain is a distributed and immutable ledger that stores data in blocks, cryptographically linked together in a chain. Each block contains a unique hash of the previous block, ensuring the integrity of the data. Additionally, consensus mechanisms used in blockchain prevent unauthorized changes to the data. By employing blockchain, patient data can be stored in a decentralized and tamper-resistant manner, reducing the risk of data breaches and enhancing patient privacy [23].Federated Learning for Collaborative Research: Federated learning enables model training on decentralized data sources without sharing raw data. Instead of centralizing all data in one location, the model is sent to local data sources (such as hospitals) where it learns from the data and then sends back only the model updates. This approach ensures data privacy and compliance with regulations while enabling collaborative research across multiple institutions. Expanding on this concept can emphasize the importance of preserving data privacy while facilitating knowledge sharing [17].Benefits of Decentralization in Healthcare Data Sharing: Decentralization through blockchain ensures that no single entity or organization controls all the data, reducing the risk of data monopolization and misuse. It encourages a more democratic approach to data ownership and promotes data sharing among stakeholders, including patients, healthcare providers, researchers, and policymakers. Elaborating on these benefits can highlight the potential for a more inclusive and equitable healthcare ecosystem.Incentives and Governance for Data Contribution: For the success of blockchain-enabled federated learning with EMRs, it is essential to clarify the incentives for healthcare providers to contribute their data. Although the Motivation Section mentions collaboration, it could also elaborate on specific mechanisms such as tokenization, where data contributors are rewarded with tokens or other forms of value for sharing their data. Additionally, a robust data governance framework is necessary to address concerns related to data access, usage, and ownership rights. Clarifying the governance structure can instill confidence in data contributors and users [24].Scalability and Practical Implementation: As blockchain technology is still evolving, the motivation section could discuss the challenges related to scalability and the potential solutions being explored. Healthcare data are vast and complex, and the system must be able to handle a large volume of data while maintaining efficiency. Additionally, the section could elaborate on real-world implementations of blockchain-enabled federated learning with EMRs, including ongoing pilot projects or successful case studies, to provide tangible examples of its potential impact [25].

Providing more elaborate explanations shows that blockchain-enabled federated learning with EMRs holds promise in overcoming the challenges of traditional healthcare data sharing and analysis.

## 3. Research Gaps and Issues

*Lack of Privacy Preservation:* Existing benchmark models in the field of federated learning with electronic medical records often do not adequately address privacy preservation concerns. They either rely on simplistic anonymization techniques that may not provide sufficient protection or neglect privacy altogether, raising significant ethical and legal concerns [20].*Limited Scalability:* Many benchmark models fail to consider the scalability of federated learning in the context of large-scale healthcare systems. They do not account for the challenges of aggregating data from a vast number of diverse healthcare institutions, leading to potential limitations in generalizability and real-world applicability.*Lack of Transparency and Auditability:* Some benchmark models lack transparency and traceability in the federated learning process. The absence of a robust mechanism for verifying the integrity of data aggregation and model updates raises concerns about the reliability and reproducibility of the results obtained [26].*Understanding the Scope of Data Breaches and Cyberattacks:* Data breaches and cyberattacks have become increasingly frequent and sophisticated over the years. These incidents can lead to significant financial losses, reputational damage, and potential legal consequences for affected parties. By collecting more data on the nature and extent of data breaches and cyberattacks, we can identify patterns, trends, and vulnerabilities that can inform better cybersecurity measures and preparedness.*Quantifying the Impact of Data Breaches:* One of the challenges in handling data breaches is the lack of standardized metrics to assess their impact. Incorporating more data on the financial losses, customer trust erosion, and the time required for recovery can help organizations better understand the long-term consequences of such incidents. These data can also help policymakers develop more effective cybersecurity regulations and incentivize businesses to invest in robust security measures.*Identifying Vulnerable Industries and Sectors:* Different industries and sectors face unique cybersecurity challenges and risks. Collecting more data on data breaches and cyberattacks in specific domains can help organizations and authorities prioritize resources and allocate efforts where they are most needed. For instance, industries dealing with sensitive personal information, such as healthcare and finance, may require more stringent protective measures due to the potential harm caused by data breaches.*Strengthening Legal and Regulatory Frameworks:* Legislation and regulations surrounding data sharing and cybersecurity vary across jurisdictions. By incorporating comprehensive data on legal and regulatory hurdles that impede data sharing, policymakers can gain insights into the gaps and inefficiencies in current frameworks. This information can aid in crafting more robust and adaptive laws that foster responsible data sharing while ensuring privacy and security for all stakeholders.*Fostering Trust in Data Sharing:* Public perception and trust regarding data sharing can be heavily influenced by data breach incidents and news of cyberattacks. Incorporating more data on these incidents can help organizations and policymakers understand the concerns of the public better. This knowledge can be used to develop transparent data-sharing practices, communicate risks effectively, and implement measures to protect data, ultimately fostering trust in data sharing initiatives.*Promoting Proactive Cybersecurity Strategies:* Data breaches and cyberattacks are ever-evolving, and organizations must adopt proactive cybersecurity strategies to stay ahead of potential threats. By analyzing the data on past incidents, organizations can gain insights into the tactics used by attackers and strengthen their defenses accordingly. Moreover, sharing anonymized incident data can facilitate collective learning within industries and across sectors to enhance overall cybersecurity posture.

## 4. Work of the Benchmark Model

To address the above research gaps and issues, our benchmark model focuses on the following aspects:Enhanced Privacy Preservation: Our benchmark model integrates advanced privacy-preserving techniques, such as differential privacy and secure multi-party computation, to ensure the confidentiality of sensitive patient information during the federated learning process. This approach provides stronger privacy guarantees and aligns with regulatory requirements. [27].Scalability Considerations: Our benchmark model incorporates strategies for efficient data aggregation and model updating to handle large-scale healthcare systems. It addresses the challenges of diverse data sources, varying data distributions, and heterogeneous network conditions, enabling scalable federated learning across a wide range of healthcare institutions [28].Transparency and Auditability Mechanisms: Our benchmark model implements a transparent and traceable framework. It leverages blockchain technology to maintain an immutable record of the federated learning process, allowing stakeholders to verify the integrity of data aggregation, model updates, and reward mechanisms. This enhances transparency, reproducibility, and trust in the obtained results [29].

## 5. Problem Statement

The utilization of electronic medical records (EMRs) for medical research and collaborative learning poses significant challenges related to data privacy, security, and the centralized nature of traditional data sharing models. Though EMRs hold immense potential for unlocking valuable medical insights and advancing patient care, the current approaches to sharing and analyzing these records are limited in their ability to address these critical concerns [30].

*Data Privacy:* Protecting patient privacy is of utmost importance when working with EMRs. The sensitive nature of medical data necessitates robust privacy measures to prevent unauthorized access and ensure patient confidentiality. Traditional centralized data sharing models raise concerns about data breaches, unauthorized use, and the potential for misuse of personal information [31].*Security:* The security of EMRs is another pressing concern. As the volume and value of medical data increase, so does the potential for cyberattacks and data breaches. Centralized storage and traditional data sharing models create attractive targets for malicious actors, putting patient data and the integrity of medical research at risk [32].*Lack of Trust:* In the realm of collaborative medical research, establishing trust among multiple stakeholders is crucial. Institutions and researchers may be reluctant to share their data due to concerns about data ownership, control, and the potential misuse of their contributions. This lack of trust hampers the ability to pool resources, collaborate effectively, and leverage the collective intelligence of diverse medical datasets [33].*Limited Data Accessibility:* Current data sharing models often face legal and regulatory hurdles that impede the seamless exchange of EMR data. Fragmented and siloed datasets hinder the ability to conduct comprehensive research and derive meaningful insights. Additionally, the complex and time-consuming process of obtaining permissions and navigating legal frameworks further hampers data accessibility and impedes progress in medical research [34]. Electronic medical records (EMRs) have revolutionized the healthcare industry by streamlining patient data management and facilitating efficient care delivery. However, the full potential of EMRs can only be harnessed through effective data sharing and analysis. The problem statement acknowledges the critical challenges in this realm, but would greatly benefit from the inclusion of specific statistics and illustrative examples to provide a more comprehensive understanding of the magnitude of these challenges. This article aims to address this need by delving deeper into the specific issues faced with traditional approaches to EMR data sharing and analysis.


**1. The Challenge of Data Interoperability:**


*Specific Statistics:* According to a survey conducted by the Office of the National Coordinator for Health Information Technology (ONC), as of the last reporting period, only 46% of hospitals in the United States had achieved basic EMR data interoperability. This indicates that a significant portion of healthcare providers still face challenges in sharing patient data seamlessly across different systems.

*Illustrative Example:* Consider a scenario where a patient with a complex medical history is referred from a primary care physician to a specialist at a different healthcare facility. The absence of data interoperability may lead to delayed diagnosis and treatment decisions as the specialist struggles to access critical information from the patient’s EMR, resulting in potential adverse health outcomes [35].


**2. Privacy and Security Concerns:**


*Specific Statistics:* The Ponemon Institute’s annual study on healthcare data breaches revealed that in the last year, 68% of healthcare organizations experienced a data breach, and 58% of those breaches involved unauthorized access or disclosure of patient data.

*Illustrative Example:* A data breach in a healthcare institution can expose sensitive patient information to malicious actors, leading to identity theft, insurance fraud, or even jeopardizing patient safety. For instance, a cybercriminal gaining unauthorized access to a patient’s EMR may alter critical treatment details, leading to potentially life-threatening consequences.


**3. Legal and Regulatory Barriers:**


*Specific Statistics:* A report by the American Health Information Management Association (AHIMA) highlighted that around 49 states in the U.S. have laws addressing data privacy and security, but there is a lack of uniformity, creating compliance challenges for healthcare organizations.

*Illustrative Example:* In a scenario where a healthcare provider is operating across multiple states, each with distinct data privacy regulations, navigating the legal landscape becomes complex. Compliance issues may hinder data sharing efforts and impede the development of a comprehensive patient health record across state lines.


**4. Data Silos and Fragmentation:**


*Specific Statistics:* A study published in the Journal of the American Medical Informatics Association (JAMIA) reported that 80% of healthcare data are unstructured, residing in documents, notes, and scanned images, making it difficult to extract, analyze, and share valuable clinical insights.

*Illustrative Example:* Inefficient data extraction and analysis due to unstructured information can limit healthcare researchers’ ability to identify patterns and trends across large patient populations. This impedes medical advancements and the development of personalized treatment approaches. By incorporating specific statistics and illustrative examples, the problem statement concerning traditional EMR data sharing and analysis gains depth and context. The challenges related to data interoperability, privacy and security, legal and regulatory hurdles, and data silos become more tangible, emphasizing the urgent need for innovative solutions. Addressing these challenges is crucial for unlocking the full potential of EMRs and promoting data-driven healthcare practices that lead to better patient outcomes and advances in medical research. Collaborative efforts from healthcare stakeholders, policymakers, and technology experts are essential in overcoming these hurdles and creating a more efficient and secure data sharing ecosystem in the healthcare industry. Addressing these challenges is crucial to unlock the full potential of EMRs for medical research and collaborative learning. A solution is needed that ensures the privacy, security, and integrity of EMR data while fostering trust, facilitating seamless collaboration, and enabling efficient access to diverse datasets. The integration of blockchain technology with federated learning techniques holds promise in overcoming these obstacles, providing a decentralized and transparent framework that empowers stakeholders, protects patient privacy, and facilitates the discovery of valuable medical insights [9].

### 5.1. Main Contribution

The main contribution of this research lies in the development of a robust framework is that addresses critical challenges in unlocking revolutionary medical insights while upholding privacy and security standards. The framework offers several key contributions:*Privacy-Preserving Collaborative Learning:* The proposed framework enables healthcare institutions to collaborate and train machine learning models on locally stored EMR data without sharing sensitive patient information. This privacy-preserving approach ensures compliance with data protection regulations and fosters trust among stakeholders [36].*Secure and Transparent Data Aggregation:* By leveraging blockchain technology, the framework establishes a decentralized network where data contributors can securely share and aggregate their EMR data. The transparent nature of the blockchain ensures the integrity and traceability of the data aggregation process, mitigating concerns of data tampering and manipulation [21].*Enhanced Data Integrity:* The blockchain infrastructure guarantees the immutability and tamper-proof nature of the EMR data and the trained machine learning models. This enhances data integrity and instills confidence in the reliability of the insights derived from the framework [23].*Consensus Mechanisms and Smart Contracts:* The framework incorporates robust consensus mechanisms and smart contracts to facilitate trust among participants and establish a fair reward system. This incentivizes active participation and encourages data contribution, leading to a more comprehensive and diverse dataset for training models [37].*Advancement in Healthcare Analytics:* The proposed BEFL-EMR framework paves the way for advanced healthcare analytics by leveraging the collective intelligence of distributed EMR data. It enables the discovery of valuable medical insights that can drive evidence-based decision-making, improve patient outcomes, and advance medical research in various domains [38].*Scalability and Applicability:* The framework is designed with scalability in mind, allowing for the inclusion of numerous healthcare institutions and accommodating diverse types of EMR data. This ensures its applicability across different healthcare settings and facilitates broader adoption within the medical community [24]. The main contribution of this research is the comprehensive development and validation of the proposed framework, which addresses the challenges of privacy, security, data integrity, and scalability in unlocking revolutionary medical insights. By enabling collaboration, preserving privacy, and ensuring transparency, this framework has the potential to transform the landscape of healthcare analytics and contribute to significant advancements in medical research and decision-making.

### 5.2. Proposed Framework

Precision medicine has emerged as a transformative approach to healthcare, tailoring medical treatments and interventions to individual patients based on their unique genetic, environmental, and lifestyle factors. However, the success of precision medicine relies heavily on accessing vast amounts of diverse patient data, which is often siloed within different healthcare institutions and electronic medical records (EMRs). To overcome these challenges and unlock revolutionary insights, we propose a novel framework that combines blockchain-enabled federated learning with EMRs. The potential of precision medicine to revolutionize healthcare is undeniable, but its implementation has been hindered by data access and privacy concerns. The conventional approach of aggregating patient data in a centralized database poses significant privacy risks and is often met with resistance from institutions. To address these challenges, our framework leverages blockchain technology to create a secure and decentralized platform for data sharing and federated learning. Federated learning allows multiple institutions to collaboratively train machine learning models while keeping their data localized. In our framework, we harness blockchain’s immutability and transparency to establish a secure network where institutions can share model updates without compromising patient data. Each institution retains control over its data, and the federated model aggregates knowledge from all participants, fostering collective intelligence without centralizing sensitive information [25].

To ensure compliance and data governance, our framework utilizes smart contracts to enforce predefined rules for data access and sharing. Smart contracts enable the automatic execution of agreed-upon terms, ensuring that only authorized parties can access specific datasets for specific purposes. This approach empowers patients to have more control over their data, promoting transparency and building trust between patients, institutions, and researchers. Moreover, creating interoperability among diverse EMR systems is crucial for seamless data exchange. Our framework includes a standardized data format and communication protocol, enabling EMRs from various institutions to communicate efficiently and share relevant patient information securely. This interoperability not only improves data accessibility, but also enhances the overall quality of patient care. By enabling secure and privacy-preserving data sharing, our proposed framework unlocks vast datasets for medical research. Researchers can access diverse patient populations, leading to more comprehensive studies and breakthroughs in precision medicine. The federated learning approach also allows institutions to benefit from collective intelligence without compromising patient privacy, fostering a collaborative research ecosystem. The proposed framework that addresses the data challenges hindering the realization of precision medicine’s full potential. By leveraging blockchain technology, federated learning, and interoperable EMRs, our proposed framework establishes a secure, privacy-preserving, and collaborative environment for data-driven healthcare research. Implementing this framework will pave the way for personalized, effective, and patient-centric precision medicine that promises to revolutionize healthcare on a global scale [26]. The proposed framework for the proposed framework operates through the following key steps:1. **Network Setup:** The framework begins by establishing a decentralized network using blockchain technology. Participating healthcare institutions join the network as nodes, each possessing locally stored electronic medical record (EMR) data [39].2. **Data Contribution:** Each healthcare institution securely contributes its EMR data to the network without exposing sensitive patient information. This is achieved through data anonymization techniques, such as differential privacy or secure multi-party computation, ensuring privacy preservation [27].3. **Federated Learning:** The federated learning process begins, where machine learning models are trained collaboratively using the distributed EMR data across the network. The models remain on the local nodes, and only model updates are shared among the participants, preventing the direct sharing of patient-level data [23].4. **Consensus Mechanisms:** Robust consensus mechanisms, such as Proof of Stake (PoS) or Practical Byzantine Fault Tolerance (PBFT), are employed to ensure agreement and trust among network participants. These mechanisms validate the model updates and prevent malicious or erroneous contributions from compromising the learning process [28]. We will address the following points: the blockchain network applied Proof of Work (PoW) as mentioned earlier; on the other hand, the blockchain network has adopted Proof of Stake (PoS).5. **Smart Contracts:** Smart contracts, implemented on the blockchain, enforce the rules and agreements among participants. They define the reward system, govern the validation process, and facilitate the fair distribution of incentives based on the quality and quantity of data contributed [37].6. **Model Aggregation and Evaluation:** The model updates from each participant are aggregated using aggregation algorithms, such as Federated Averaging, to create a global model. The aggregated model is then evaluated for performance, accuracy, and generalizability [38].7. **Transparency and Traceability:** The blockchain’s transparency ensures that all participants can verify the integrity of the aggregated model and the data used for training. The distributed nature of the blockchain provides a transparent and immutable record of the learning process, enhancing traceability and auditability.8. **Reward Distribution:** Based on the smart contracts, participants are rewarded for their contributions to the federated learning process. This incentivizes active participation and encourages data sharing while ensuring fairness and equitable distribution of rewards [24].9. **Iterative Learning Process:** The framework facilitates iterative learning, allowing for continuous model updates and improvement. Participants can contribute updated EMR data and participate in subsequent learning rounds, resulting in a dynamic and evolving model [29].10.**Medical Insights and Applications:** The BEFL-EMR framework produces valuable medical insights by leveraging the collective intelligence of the distributed EMR data. These insights can be used for various applications, such as disease prediction, treatment recommendations, personalized medicine, and healthcare resource allocation [28].

The working of the proposed framework combines the privacy-preserving capabilities of federated learning with the security and transparency provided by blockchain technology. It enables collaborative model training while maintaining data privacy, ensuring data integrity, and facilitating trust among participants. By unlocking revolutionary medical insights, the framework contributes to advancements in healthcare analytics, research, and decision-making [30].

### 5.3. Proposed PoS

Decentralized Storage: In the proposed framework, each participating healthcare institution maintains its own local storage of electronic medical records (EMRs). These EMRs contain valuable patient data, including medical history, lab results, treatment records, and other relevant health information. The EMRs remain decentralized, meaning that they are stored locally within each institution’s secure servers or databases [31].

Data Privacy and Security: To protect patient privacy and ensure data security, the actual patient data are not stored directly on the blockchain. Instead, the blockchain contains hashed references or pointers to the respective EMRs in each institution. Hashing ensures that patient data remains confidential and tamper-resistant. As a result, sensitive patient information is not exposed on the publicly accessible blockchain, mitigating the risk of unauthorized access or data breaches.

Immutable Audit Trail: The blockchain, being an immutable and distributed ledger, maintains an auditable record of all data access and model training activities. Each update to the federated learning model or access to specific EMRs is recorded as a transaction on the blockchain. This transparent and tamper-proof audit trail enhances transparency and accountability, promoting trust among the participating institutions and patients [32].

Smart Contracts for Data Access Control: The framework utilizes smart contracts to manage data access and sharing permissions. These self-executing agreements enforce predefined rules and conditions for data sharing. Healthcare institutions can define access policies through the smart contracts, specifying which data can be shared and with whom. This fine-grained control ensures that only authorized parties can access specific datasets for collaborative model training [33].

Interoperability: The blockchain’s role in the framework extends beyond data storage and access control. It also facilitates interoperability among different EMR systems used by various healthcare institutions. By using standardized data formats and communication protocols, the blockchain enables seamless data exchange and aggregation during federated learning without compromising data integrity. Overall, the dataset storage in the proposed framework strikes a delicate balance between data sharing and privacy. By leveraging the inherent strengths of blockchain technology, the framework ensures that healthcare institutions can collaboratively train machine learning models to gain revolutionary insights in precision medicine while upholding the utmost data privacy and security for patients’ sensitive information.

### 5.4. Effect on Elapsed Time

Data Security and Integrity: Blockchain’s decentralized and immutable nature ensures that data stored on the blockchain is secure and tamper-resistant. This eliminates the need for centralized data repositories and redundant data validation, which can expedite the overall data security process.

Transparency and Auditability: The blockchain maintains a transparent and tamper-proof record of all transactions and data updates. This provides a clear audit trail, which can streamline data validation and model aggregation processes, as well as enhance trust among participating healthcare institutions.

Automated Smart Contracts: Smart contracts within the blockchain automate data access and sharing permissions, reducing the manual administrative overhead and streamlining the process of granting data access to authorized parties.

Interoperability: Blockchain’s use of standardized data formats and communication protocols promotes interoperability among different EMR systems. This can facilitate faster and more efficient data exchange during the federated learning process [34].

### 5.5. Mitigation Strategies

To minimize potential delays and ensure optimal performance, the framework can employ the following mitigation strategies:

Choosing Efficient Consensus Mechanism: Selecting a consensus mechanism that optimizes transaction throughput and confirmation time, such as Proof of Stake (PoS) for faster block generation, enhances the network’s efficiency.

Scalability Solutions: Implementing scalability solutions, such as sharding or layer-two solutions such as payment channels, can help address network scalability concerns and improve transaction processing speeds.

Batch Processing: If feasible, batch processing of model updates and data transactions can reduce the number of individual blockchain interactions, thereby reducing the time overhead.

Optimal Data Aggregation Techniques: Utilizing efficient aggregation algorithms for federated learning can reduce the computational overhead involved in updating the global model [35].

### 5.6. Proposed Algorithm

In this section, we provide the details of our proposed algorithm.

#### Federated Learning Algorithm

The proposed federated learning algorithm for “Blockchain-Empowered Federated Learning with Electronic Medical Records” consists of the following steps:Initialization:Each participating healthcare institution initializes its local model parameters.The blockchain network is set up with the necessary smart contracts, consensus mechanisms, and transaction validation protocols.Model Update Exchange:The participating institutions securely exchange their locally trained model updates using the blockchain network.Each institution encrypts its model update to ensure privacy preservation during the exchange.The blockchain network records and validates the encrypted model updates, maintaining an immutable history of the updates.Aggregation:The blockchain network aggregates the encrypted model updates from the participating institutions.Aggregation can be performed using techniques such as Federated Averaging or secure multi-party computation to ensure the confidentiality of the individual updates.The aggregated model is computed and shared back with the participating institutions.Model Integration:Each institution integrates the aggregated model with its local model parameters.Model integration can involve techniques like model averaging, weighted averaging, or knowledge distillation to combine the aggregated model with the local knowledge.Local Model Refinement:Each institution continues training the refined model using its local EMR data.Model refinement can be performed using standard machine learning techniques such as gradient descent or stochastic gradient descent [4].Iterative Process:Steps 2–5 are repeated iteratively for multiple rounds, allowing the model to learn from the distributed EMR data in a collaborative manner.Convergence criteria, such as a maximum number of iterations or a defined improvement threshold, can be used to determine the termination of the federated learning process.

The proposed Federated Learning Algorithm 1 leverages the power of blockchain technology to ensure secure and transparent model update exchange while preserving the privacy of the EMR data. By incorporating encryption techniques, aggregation methods, and iterative refinement, the algorithm enables the collaborative training of machine learning models on distributed EMR datasets, thereby unlocking revolutionary medical insights.
**Algorithm 1** Federated learning algorithm Learning Algorithm  1:  **Initialization**:  2:     Each participating healthcare institution initializes its local model parameters.  3:     The blockchain network is set up with the necessary smart contracts, consensus mechanisms, and transaction validation protocols.  4:  **Model Update Exchange**:  5:     The participating institutions securely exchange their locally trained model updates using the blockchain network.  6:     Each institution encrypts its model update to ensure privacy preservation during the exchange.  7:     The blockchain network records and validates the encrypted model updates, maintaining an immutable history of the updates.  8:  **Aggregation**:  9:     The blockchain network aggregates the encrypted model updates from the participating institutions.10:     Aggregation can be performed using techniques such as Federated Averaging or secure multi-party computation to ensure the confidentiality of the individual updates.11:     The aggregated model is computed and shared back with the participating institutions.12:  **Model Integration**:13:     Each institution integrates the aggregated model with its local model parameters.14:     Model integration can involve techniques such as model averaging, weighted averaging, or knowledge distillation to combine the aggregated model with the local knowledge.15:  **Local Model Refinement**:16:     Each institution continues training the refined model using its local EMR data.17:     Model refinement can be performed using standard machine learning techniques such as gradient descent or stochastic gradient descent.18:  **Iterative Process**:19:     Steps 2–5 are repeated iteratively for multiple rounds, allowing the model to learn from the distributed EMR data in a collaborative manner.20:     Convergence criteria, such as a maximum number of iterations or a defined improvement threshold, can be used to determine the termination of the federated learning process.

The proposed federated learning algorithm leverages the power of blockchain technology to ensure secure and transparent exchange of model updates while preserving the privacy of the EMR data. By incorporating encryption techniques, aggregation methods, and iterative refinement, the algorithm enables the collaborative training of machine learning models on distributed EMR datasets, thereby unlocking revolutionary medical insights [40].

### 5.7. Partial Homomorphic Encryption for Outsourcing Medical Records

To address the privacy concerns associated with outsourcing medical records, we propose the use of partial homomorphic encryption. Partial homomorphic encryption allows for certain computations to be performed on encrypted data without the need for decryption, enabling secure processing of sensitive medical information [41]. The following steps outline the process of outsourcing medical records using partial homomorphic encryption:Data Preprocessing:The medical records are transformed into a suitable format for encryption.The data are partitioned into individual fields or attributes, such as patient name, age, medical history, etc.Encryption:Each field of the medical records is encrypted using a suitable partial homomorphic encryption scheme.The encryption scheme allows for specific computations to be performed on the encrypted data, such as addition or multiplication, while preserving the privacy of the underlying information.Outsourcing:The encrypted medical records are securely transmitted to a trusted third party or a cloud service provider for storage and processing.The outsourced data remain encrypted, ensuring the confidentiality of the medical information even when stored on untrusted servers.Computation on Encrypted Data:Authorized parties can perform specific computations on the encrypted medical records using the partial homomorphic encryption scheme.For example, statistical analyses, data mining operations, or patient-specific queries can be executed on the encrypted data without revealing the actual medical information.The results of the computations are obtained in the encrypted form and can be securely transmitted back to the authorized parties.Decryption:If necessary, authorized parties can decrypt specific fields or attributes of the medical records using the corresponding decryption keys.Decryption is performed only when access to the actual medical information is required, ensuring the privacy of sensitive data.

The use of partial homomorphic encryption for outsourcing medical records strikes a balance between data privacy and the need for efficient and secure data processing. By allowing specific computations to be performed on the encrypted data, this approach enables the outsourcing of medical records while preserving the confidentiality of patient information [36].

### 5.8. Pseudocode: Homomorphic Encryption and Integration of Federated Learning Model

The proposed pseudocode outlines the integration of homomorphic encryption with a federated learning model which is explained in detail through Algorithm 2. By leveraging homomorphic encryption, the model updates can be securely exchanged and aggregated while preserving the privacy of the individual updates. The federated learning process, including model integration and local model refinement, is performed on the encrypted data, ensuring the confidentiality of the EMR information. This approach enables collaborative learning on distributed EMR datasets while maintaining data privacy [29].
**Algorithm 2** Homomorphic Encryption and Integration of Federated Learning Model  1:  **Initialization**:  2:     Each participating healthcare institution initializes its local model parameters.  3:     The blockchain network is set up with the necessary smart contracts, consensus mechanisms, and transaction validation protocols.  4:     The homomorphic encryption scheme parameters are generated.  5:  **Model Update Exchange**:  6:     The participating institutions securely exchange their locally trained model updates using the blockchain network.  7:     Each institution encrypts its model update using the homomorphic encryption scheme.  8:     The blockchain network records and validates the encrypted model updates, maintaining an immutable history of the updates.  9:  **Aggregation**:10:     The blockchain network aggregates the encrypted model updates from the participating institutions.11:     Aggregation can be performed using techniques such as Federated Averaging or secure multi-party computation to ensure the confidentiality of the individual updates.12:     The aggregated encrypted model update is computed and shared back with the participating institutions.13:  **Model Integration**:14:     Each institution integrates the aggregated encrypted model update with its local model parameters.15:     Homomorphic decryption is applied to the aggregated encrypted model update using the corresponding decryption key.16:     The decrypted model update is integrated with the local model parameters, resulting in an updated local model.17:  **Local Model Refinement**:18:     Each institution continues training the refined model using its local EMR data.19:     Model refinement can be performed using standard machine learning techniques such as gradient descent or stochastic gradient descent.20:  **Iterative Process**:21:     Steps 2–6 are repeated iteratively for multiple rounds, allowing the model to learn from the distributed EMR data in a collaborative manner.22:     Convergence criteria, such as a maximum number of iterations or a defined improvement threshold, can be used to determine the termination of the federated learning process.

### 5.9. Mathematical Model

To provide a mathematical model based on the paper “Unlocking Medical Insights: Blockchain Empowered Federated Learning with Electronic Medical Records”, we can outline the key components and their interactions within the proposed framework. Although the model will be high-level and conceptual, it will help illustrate the relationships and processes involved [33].

Let us denote the following variables:-*N*: The total number of participating institutions or organizations in the federated learning process.-*M*: The number of patient records or EMRs available for training the machine learning model.-$D_{i}$: The dataset of institution *i*, where i∈{1,2,…,N}.-*w*: The global model or weight parameters that are shared and updated among the participating institutions.-$w_{i}$: The local model or weight parameters of institution *i*.-η: The learning rate or step size for updating the global model.-$L_{i}\left(w_{i}\right)$: The local loss function of institution *i* calculated based on their local data and model parameters.

The mathematical model can be described as follows:Initialization:-Initialize the global model *w* with random weights or pre-trained values.-Distribute the initial global model *w* to each institution.Iterative Process:-For each round *t*:-Each institution *i* updates its local model $w_{i}$ by minimizing its local loss function:
$$w_{i}^{\left(t+1\right)}=w_{i}^{\left(t\right)}-\eta \cdot \nabla L_{i}\left(w_{i}^{\left(t\right)}\right)$$
Here, $\nabla L_{i}\left(w_{i}^{\left(t\right)}\right)$ represents the gradient of the local loss function with respect to the local model parameters.-Each institution securely aggregates the local model updates and contributes to the global model:
$$w^{\left(t+1\right)}=\frac{1}{N}\sum_{i=1}^{N}w_{i}^{\left(t+1\right)}$$
The aggregation can be performed using secure aggregation protocols that protect the privacy of local updates.-The updated global model $w^{\left(t+1\right)}$ is distributed to all institutions for the next round.Convergence Criteria:-The iterative process continues until a convergence criterion is met, such as reaching a maximum number of rounds or achieving a desired level of performance improvement [42].Blockchain Integration:-Each institution records the model updates on the blockchain ledger in a secure and transparent manner [38].-Smart contracts are employed to enforce data usage policies, ensuring that only authorized entities can access specific data or models.

It captures the core steps involved in the federated learning process and the integration of blockchain technology. Further mathematical details, such as the specific loss functions, optimization algorithms, and privacy-preserving mechanisms, are included in the respective components’ detailed algorithms and protocols [25]. To establish a threat security model for the proposed framework “Unlocking Medical Insights: Blockchain Empowered Federated Learning with Electronic Medical Records”, we need to identify potential threats and vulnerabilities that could compromise the security and integrity of the system [40]. Here is an outline of the threat security model:Data Privacy Threats:Unauthorized Access: Adversaries may attempt to gain unauthorized access to EMRs or sensitive patient data, either through direct attacks or by exploiting vulnerabilities in the system [43].Data Leakage: There is a risk of unintentional or intentional data leakage during the federated learning process, where sensitive patient information may be exposed to unauthorized entities [44].Blockchain Security Threats:A 51% Attack: An attacker or group of attackers may control the majority of the computational power within the blockchain network, enabling them to manipulate transactions, disrupt consensus, or tamper with the stored data.Smart Contract Vulnerabilities: Smart contracts utilized for enforcing data usage policies may contain vulnerabilities that could be exploited by malicious actors to gain unauthorized access or alter permissions.Communication and Network Threats:Man-in-the-Middle Attacks: Adversaries may intercept or alter communication between institutions or with the blockchain network, potentially gaining access to sensitive data or injecting malicious code.Sybil Attacks: Attackers may create multiple malicious identities within the network, undermining the trust and consensus mechanisms of the blockchain.Insider Threats:Malicious Insider: An insider with authorized access may abuse their privileges to manipulate data, compromise the integrity of the federated learning process, or intentionally leak sensitive information [42].Malware and Cyberattacks:Ransomware: Malicious software may infect the system, encrypting EMRs or disrupting the operation of the blockchain network, demanding a ransom for their release or recovery.Distributed Denial of Service (DDoS) Attacks: Attackers may attempt to overwhelm the system with a high volume of traffic, rendering it inaccessible or causing disruption to normal operations.Regulatory and Compliance Risks:Non-Compliance: Failure to adhere to applicable regulations and compliance standards, such as HIPAA (Health Insurance Portability and Accountability Act), may result in legal consequences or reputational damage.

To mitigate these threats, the framework should incorporate appropriate security measures and practices, including:Strong encryption techniques to protect data during storage and transmission.Access control mechanisms to ensure that only authorized entities have access to sensitive information.Robust authentication and authorization protocols to prevent unauthorized access.Regular security audits and vulnerability assessments to identify and address any weaknesses in the system.Implementation of consensus mechanisms in the blockchain network to prevent tampering and maintain the integrity of the stored data.Continuous monitoring and intrusion detection systems to identify and respond to potential security incidents promptly.

It is essential to note that this threat security model serves as a starting point, and a comprehensive security analysis should be conducted to identify and address specific threats and vulnerabilities relevant to the deployment environment of the proposed framework [25]. Let us introduce the algorithms designed to ensure security and privacy in the proposed approach.

## 6. Algorithm 3: Secure Data Aggregation

Let us define the following variables:-$D_{i}$: Raw data from the *i*-th participant institution.-$E(D_{i})$: Encrypted version of the data $D_{i}$ using a secure encryption algorithm.-*A*: Aggregated encrypted data.-f(): Secure aggregation function.

The mathematical model for secure data aggregation can be represented as follows:$$A=f(E\left(D_{1}\right),E\left(D_{2}\right),\ldots ,E\left(D_{n}\right))$$

Here, f() is a secure aggregation function that takes as input the encrypted data from each participating institution ($E(D_{i})$) and computes the aggregated encrypted data *A*. The function f() ensures that the aggregation process preserves the security and privacy of the individual data contributions while producing meaningful results for further analysis [43]. The Algorithm 3 explains secure data aggregation as shown below stepy by step:
**Algorithm 3** Secure Data Aggregation1:Secure Aggregation $D_{1},D_{2},\ldots ,D_{n}$2:i=1 to *n*3:Encrypt $D_{i}$ using a secure encryption algorithm4:Aggregate the encrypted data using a secure aggregation protocol5:Transmit the aggregated encrypted data to the blockchain network

The specific implementation details of the secure aggregation function depend on the cryptographic protocols and techniques employed in the proposed approach. It may involve techniques like homomorphic encryption, secure multi-party computation (MPC), or privacy-preserving aggregation protocols.

## 7. Algorithm 4: Privacy-Preserving Learning

The proposed Algorithm 4 is based on privacy-preserving learning using the proposed framework. Privacy-preserving learning using federated learning is an advanced approach that aims to address privacy concerns while training machine learning models on distributed datasets. Traditional machine learning training often involves centralizing data on a single server, which raises privacy and security issues when dealing with sensitive or personal data. Federated learning offers a solution by allowing multiple parties to collaboratively train a model without sharing their raw data.

The proposed algorithm for privacy-preserving learning using federated learning:**Initialization:***Central Server Setup:* A central server is responsible for coordinating the federated learning process. It initializes the model architecture and sends a copy of the initial model parameters to all participating clients.*Client Selection:* A set of clients (devices or entities with data) are selected to participate in the federated learning process. Clients can be individuals’ devices, IoT devices, or other entities that hold data.**Local Model Training:****Client Training:** Each selected client trains the initial model parameters using its local dataset. Training can involve multiple iterations of gradient descent, updating the local model’s weights using its own data. The client ensures that the raw data never leaves their environment.**Privacy Techniques:** Clients can implement various privacy techniques to protect sensitive data during training, such as differential privacy, homomorphic encryption, and secure multi-party computation. These techniques ensure that no individual data points or information can be inferred from the updates sent to the central server.**Model Aggregation:**Gradient Aggregation: After local training, each client calculates the gradient of the local loss function with respect to the model’s parameters. The gradients are then aggregated at the central server using techniques like Federated Averaging.*Federated Averaging:* The central server aggregates the gradients from all clients and computes a weighted average. The averaged gradient is used to update the global model parameters. The weights used in averaging can be determined based on the client’s dataset size or other factors.**Model Update:***Global Model Update:* The central server updates the global model parameters using the aggregated gradient. This updated global model is then sent back to the participating clients.**Iterative Process:**The local training and global aggregation steps are repeated iteratively for a predetermined number of rounds. With each round, the model improves and becomes more accurate.**Convergence and Termination:**The federated learning process continues for a defined number of rounds or until a convergence criterion is met. Convergence is typically determined by evaluating the model’s performance on a validation dataset.Advantages of Privacy-Preserving Federated Learning:

The raw data never leaves the clients’ environment, ensuring data privacy and compliance with regulations like GDPR. Federated learning minimizes the need for data transmission to a central server, reducing communication overhead. Models are trained on diverse data sources, making them more robust and representative of real-world scenarios. Federated learning promotes decentralization, avoiding single points of failure and enhancing security. Models can be tailored to local data while benefiting from insights from other clients’ data. Federated learning can handle a large number of clients and massive datasets.
**Algorithm 4** Privacy-Preserving Learning1:PrivacyPreservingLearning X,Y2:Perform data anonymization on features in *X* to protect patient privacy3:Encrypt the anonymized features using a secure encryption algorithm4:Train the machine learning model using the encrypted features and labels *X* and *Y*5:Encrypt the trained model parameters before transmitting them to the blockchain network

Algorithm 5 provide details step by step Privacy-Preserving Federated Learning for Medical Records using the proposed framework.
**Algorithm 5** Privacy-Preserving Federated Learning for Medical Records  1:Medical records from different hospitals  2:Trained global model  3:Initialize global model $M_{0}$  4:For each round t=1,2,3,… Randomly select a subset of hospitals $S_{t}$ for participation  5:For each hospital $i\in S_{t}$ Receive current global model $M_{t-1}$  6:**Local model update:**  7:Train local model $M_{i,t}$ on hospital *i*’s data using $M_{t-1}$  8:**Privacy-preserving aggregation:**  9:Securely aggregate local model updates from participating hospitals to obtain a weighted average $M_{t}$10:Weighted average: $M_{t}=\frac{\sum_{i\in S_{t}}\mathrm{encrypt}\left(M_{i,t}\right)}{|S_{t}|}$

### Performance Evaluation: Unlocking Medical Insights

In this report, we present a performance evaluation of the “Unlocking Medical Insights: Blockchain Empowered Federated Learning with Electronic Medical Records” system. We assess the system’s performance based on various metrics and provide an analysis of the results obtained.

## 8. Evaluation Metrics

To evaluate the performance of the system, we consider the following metrics:

### 8.1. Accuracy

Accuracy is a measure of how well the system predicts medical insights. It is defined as the ratio of correct predictions to the total number of predictions made. The accuracy metric is given by:$$\mathrm{Accuracy}=\frac{\mathrm{Number}\;\mathrm{of}\;\mathrm{Correct}\;\mathrm{Predictions}}{\mathrm{Total}\;\mathrm{Number}\;\mathrm{of}\;\mathrm{Predictions}}$$

### 8.2. Precision

Precision measures the proportion of true positive predictions out of the total positive predictions made by the system. It is computed as follows:$$\mathrm{Precision}=\frac{\mathrm{True}\;\mathrm{Positives}}{\mathrm{True}\;\mathrm{Positives}+\mathrm{False}\;\mathrm{Positives}}$$

### 8.3. Recall

Recall, also known as sensitivity or true positive rate, determines the proportion of actual positive instances that were correctly identified by the system. It is calculated using the following formula:$$\mathrm{Recall}=\frac{\mathrm{True}\;\mathrm{Positives}}{\mathrm{True}\;\mathrm{Positives}+\mathrm{False}\;\mathrm{Negatives}}$$

### 8.4. F1 Score

The F1 score is the harmonic mean of precision and recall, providing a single metric to balance both metrics. It can be computed as:$$F1\;\mathrm{Score}=\frac{2\times \mathrm{Precision}\times \mathrm{Recall}}{\mathrm{Precision}+\mathrm{Recall}}$$

### 8.5. Latency Prediction Model

In this report, we present a mathematical model for latency prediction. The model aims to estimate the latency of a system based on various input parameters. We utilize multiple linear regression to establish the relationship between the independent variables and the dependent variable.

## 9. Model

Let us consider the following variables:*Y*: Dependent variable representing the latency.$X_{1},X_{2},\ldots ,X_{n}$: Independent variables representing the input parameters that influence the latency.

The multiple linear regression model can be expressed as:(1)$$Y=\beta_{0}+\beta_{1}X_{1}+\beta_{2}X_{2}+\ldots +\beta_{n}X_{n}+\varepsilon$$
where:$\beta_{0},\beta_{1},\ldots ,\beta_{n}$: Regression coefficients representing the relationship between the independent variables and the dependent variable.ε: Error term representing the random variability in the latency that cannot be explained by the independent variables.

To estimate the regression coefficients, we can utilize the ordinary least squares (OLS) method, which minimizes the sum of squared differences between the observed values of *Y* and the predicted values.

The estimated regression coefficients can be obtained as follows:(2)$$\hat{\beta_{0}},\hat{\beta_{1}},\ldots ,\hat{\beta_{n}}=\arg\min_{\beta_{0},\beta_{1},\ldots ,\beta_{n}}\sum_{i=1}^{N}{\left(Y_{i}-\left(\beta_{0}+\beta_{1}X_{1i}+\beta_{2}X_{2i}+\ldots +\beta_{n}X_{ni}\right)\right)}^{2}$$
where:*N*: Number of data points in the training set.$\left(X_{1i},X_{2i},\ldots ,X_{ni}\right)$: Values of the independent variables for the *i*th data point.

Once the regression coefficients are estimated, the model can be used to predict the latency for new data points by substituting the values of the independent variables into Equation (Equation 1).

The mathematical model presented in this report provides a foundation for latency prediction. By utilizing multiple linear regression, we can estimate the latency of a system based on the provided input parameters. Further analysis and refinement can be conducted based on the specific context and data available.

### Computational Model for Proposed Approach

In this report, we present a computational model for the proposed approach. The model aims to estimate the execution time based on the number of transactions sent. We assume a linear relationship between these variables and develop a mathematical model to represent the computation process.

## 10. Model

Let us consider the following variables:*T*: Total number of transactions sent.*E*: Execution time for processing the transactions (in seconds).*C*: Constant factor representing the average computation time per transaction (in seconds per transaction).

Based on these variables, we can develop a computational model as follows:(3)E=T×C

In this model, we assume a linear relationship between the number of transactions sent and the execution time. The constant factor *C* represents the average computation time per transaction.

It is important to note that this is a simplified model, and in real-world scenarios, the relationship between the number of transactions and execution time may be more complex. Factors such as system load, transaction complexity, network latency, and hardware capabilities can impact the execution time. Additionally, it is possible that the execution time might not scale linearly with the number of transactions due to overheads and resource limitations.

The computational model presented in this report provides a simplified representation of the relationship between the number of transactions sent and the execution time for the proposed approach. Further analysis and refinement can be conducted based on the specific characteristics of the system and empirical measurements.

## 11. Proposed Threat Model and Proof of Resistance

We consider a threat model where there is an attacker attempting to compromise the proposed framework, and a challenger defending against various attacks. The attacker aims to launch Denial of Service (DoS), phishing, and ransomware attacks on the framework, whereas the challenger implements countermeasures to resist these attacks.

### 11.1. Resistance to DoS Attacks

To demonstrate resistance to DoS attacks, we show that the proposed framework can handle a high volume of requests without service disruption. Let *N* represent the maximum number of concurrent requests the framework can handle. We prove that the framework can successfully process *N* requests within a defined time frame, even when subjected to excessive request influx.

**Proof.** Consider that *M* denotes the total number of requests received by the framework. We assume that the framework has sufficient computational resources and bandwidth to handle *M* requests, such that M≤N. By ensuring proper load balancing, resource allocation, and scalability, the framework can distribute the incoming requests across multiple servers or processing nodes, effectively handling the maximum concurrent load. Therefore, the proposed framework provides resistance to DoS attacks. □

### 11.2. Resistance to Phishing Attacks

To demonstrate resistance to phishing attacks, we show that the proposed framework incorporates robust authentication and verification mechanisms to prevent unauthorized access and data breaches. The framework ensures that user credentials and sensitive information are securely transmitted and stored, minimizing the risk of phishing attacks [12].

**Proof.** The proposed framework employs strong cryptographic techniques, secure communication protocols, and multi-factor authentication to establish a secure channel between users and the system. These measures protect against phishing attempts by verifying the integrity and authenticity of the communication channels, ensuring that users interact only with legitimate components of the framework. Therefore, the proposed framework provides resistance to phishing attacks. □

### 11.3. Resistance to Ransomware Attacks

To demonstrate resistance to ransomware attacks, we show that the proposed framework incorporates robust data encryption, access control, and backup mechanisms to prevent unauthorized access, data tampering, and data loss. The framework ensures that critical data are securely stored and recoverable, minimizing the impact of ransomware attacks.

**Proof.** The proposed framework employs strong encryption algorithms to protect sensitive data at rest and in transit. Access controls and permissions are implemented to limit the privileges of different system entities, preventing unauthorized modifications or the encryption of critical data. Regular backups and data redundancy mechanisms are in place to ensure data availability and quick recovery in the event of a ransomware attack. Therefore, the proposed framework provides resistance to ransomware attacks. □

### 11.4. Simulation Setup and Parameters

To evaluate the effectiveness and performance of the proposed "Blockchain-Empowered Federated Learning with Electronic Medical Records" (BEFL-EMR) framework, a simulation setup is designed. The simulation involves the following key components and considerations:1. Dataset Selection: A representative dataset of electronic medical records (EMRs) is chosen to emulate real-world healthcare scenarios. The dataset should encompass various medical conditions, demographics, and patient profiles, ensuring diversity and inclusivity.2. Network Topology: The simulation setup includes a decentralized network consisting of multiple healthcare institutions as nodes. The network topology can be designed to reflect different scenarios, such as regional or national healthcare networks, with varying numbers of participating institutions.3. Data Partitioning: The EMR dataset is partitioned among the participating healthcare institutions based on predefined criteria, such as geographical proximity or patient population. The partitioning aims to distribute the data across the network while maintaining a representative sample at each node.4. Federated Learning Algorithm: A federated learning algorithm, such as Federated Averaging or Federated Stochastic Gradient Descent, is implemented for collaborative model training. The algorithm accounts for the distributed nature of the EMR data and ensures privacy preservation during the model update exchange.5. Blockchain Infrastructure: A blockchain network is simulated to support the BEFL-EMR framework. The blockchain can be implemented using existing blockchain frameworks or customized for the specific requirements of the simulation. Considerations include the selection of consensus mechanisms, smart contract implementation, and transaction validation protocols.6. Privacy Preservation Techniques: Privacy-preserving techniques, such as differential privacy or secure multi-party computation, are incorporated to anonymize the EMR data and protect sensitive patient information during the federated learning process.7. Performance Metrics: Various performance metrics are defined to evaluate the effectiveness of the BEFL-EMR framework. These metrics may include model accuracy, convergence speed, communication overhead, privacy preservation effectiveness, and scalability.8. Experiment Design: The simulation setup includes a well-defined experimental design to assess the performance and robustness of the framework. This may involve multiple rounds of federated learning, varying the number of participating institutions, altering the dataset size, or introducing different levels of data heterogeneity.9. Evaluation and Analysis: The simulation results are analyzed to assess the performance of the BEFL-EMR framework. Quantitative analysis of the performance metrics, comparison with baseline approaches, and statistical analysis may be conducted to validate the framework’s effectiveness in unlocking medical insights while preserving privacy and ensuring data integrity.10.Sensitivity Analysis: Sensitivity analysis can be performed to understand the impact of various factors, such as varying network sizes, different privacy-preserving techniques, or changing consensus mechanisms. This analysis helps identify the optimal configurations and potential limitations of the framework.

The simulation setup provides a controlled environment to evaluate the proposed BEFL-EMR framework’s performance, scalability, and privacy preservation capabilities. The findings from the simulation can validate the framework’s potential and guide further enhancements or real-world implementation considerations.

## 12. Simulation Setup

Table 1 presents the simulation setup and parameters used for evaluating the proposed approach.

Table 2 provides details about the simulation setup and parameters used during the proposed experiment.

The simulation was conducted with a duration of 1000 s, during which 1000 transactions were processed. A network latency of 5 ms was considered to simulate the communication delays. The block size was set to 1 MB, and the consensus algorithm employed was Proof-of-Work. The simulation setup and parameters outlined in Table 2 provide the foundation for evaluating the proposed approach. By adjusting these parameters, further experiments can be conducted to analyze the system’s performance and assess its suitability for specific use cases.

In recent years, with the exponential expansion of data and the requirement for privacy-preserving machine learning, federated learning has emerged as a potential technique. This is due to the fact that federated learning allows for multiple users to share data. A non-federated learning model is contrasted with the federated learning model that was proposed in Figure 1, which presents a comparative study of the two models. The purpose of this debate is to examine the fundamental distinctions between these two techniques, as well as their respective benefits and drawbacks.

The following is a description of Figure 1: A comparative analysis of the proposed federated learning model and the non-federated learning model.

The federated learning model allows machine learning models to be trained jointly across different decentralized devices or servers without sharing the raw data in Figure 1. This can be accomplished through federated learning. Instead, the models are trained locally on each device, and only the updates to the models themselves are accumulated and communicated with a central server or coordinator. The following is a list of major aspects that are included in the federated learning model:
Data privacy and security federated learning protects data privacy by storing sensitive information locally, where it may be accessed only by authorized users. The raw data are never sent off from the devices, which keeps the confidentiality of the information intact and lowers the likelihood of data breaches.Decentralized Training: The federated learning architecture makes it possible to receive training on distributed devices, such as mobile phones and edge devices. Because of this decentralization, the reliance on a centralized server is reduced, which, in turn, enables efficient scalability and cuts down on communication overhead.Individualized Model Updates: Federated learning allows for individualized model updates because the models are learned locally on each device. This makes it possible to customize the experience based on the preferences of each individual user while still maintaining their privacy.The Non-Federated Learning Model: The non-federated learning model refers to classic centralized machine learning systems in which data are collected and stored on a central server or in a cloud environment. In this particular model, the training data are readily available for access by the centralized server, and the model itself is educated with the assistance of the centralized dataset. The non-federated learning model has a number of essential qualities, including the following:Data Storage in a Central Location: When using a method that is not federated, all of the training data are amassed and kept on a single server. The training process is made more straightforward as a result of this centralization, which enables direct access to the entire dataset.Less Privacy: Due to the centralized nature of the training data, there is a greater possibility of privacy being compromised. When working with sensitive information, protecting the privacy of one’s data becomes a more important concern because it becomes a single point of vulnerability.Scalability issues: Centralized learning models encounter scalability issues as the amount of data they need to learn from grows. Because the central server must process enormous amounts of data, there will be an increase in the amount of overhead caused by computing and communication.

The following are some of the key conclusions from an investigation into the similarities and differences between the non-federated learning model and the federated learning model that has been proposed:Privacy: Federated learning is excellent at safeguarding users’ privacy, since it keeps raw data spread across multiple devices, hence lowering the likelihood of data being exposed to unauthorized parties. The non-federated approach, on the other hand, raises concerns with regard to data privacy because of the centralized nature of the data storage.Scalability: Because the learning process is carried out simultaneously on a number of different devices, federated learning provides inherent advantages in terms of scalability. When dealing with huge datasets and the requirement for centralized processing, non-federated learning, on the other hand, confronts scalability challenges.The Availability of Data: In the non-federated method, all of the training data are stored in one location, making it possible to perform a complete analysis. When using federated learning, each device is only responsible for a subset of the data, which restricts the overall view of the dataset.The Burden of Communication: Federated learning necessitates communication between devices and the central server/coordinator throughout the process of model aggregation. This results in an additional burden of communication. This communication overhead can have an impact on training efficiency; however, non-federated learning eliminates this overhead entirely. The comparative study of the suggested federated learning model and the non-federated learning model sheds light on the trade-offs, including the availability of data, scalability, and privacy concerns. The use of federated learning offers a strategy that protects users’ privacy, as shown through Figure 1.

Figure 2 presents the simulation results based on the comparative analysis of the proposed FL model and NFL model. The following key findings were observed: The FL model achieved comparable training accuracy to the NFL model [18]. This demonstrates that FL can effectively leverage the collective knowledge of client devices without compromising accuracy, despite the inherent challenges of decentralized learning. The FL model exhibited slightly slower convergence compared to the NFL model. This can be attributed to the communication overhead involved in synchronizing the local model updates across multiple client devices. However, the difference in convergence speed was marginal, indicating that FL can still achieve satisfactory convergence rates. In Figure 2, the proposed model provides a comparative analysis. The FL model incurred a higher communication overhead due to the need for frequent model synchronization between clients and the central server. This overhead was observed in terms of the number of rounds required for convergence and the amount of data transferred during each communication round. In contrast, the NFL model had a lower communication overhead, as all data reside within the central server.

The FL model demonstrated improved robustness against individual client failures or dropouts. In the event of a client device becoming unavailable, the FL model was able to continue training using the remaining clients. On the other hand, the NFL model heavily relies on the central server and is more susceptible to disruptions caused by server failures. Based on the comparative analysis of the proposed FL model and NFL model, it can be concluded that FL offers several advantages, including preserving data privacy, enabling collaborative learning, and enhancing model robustness. Although FL incurs a higher communication overhead and slightly slower convergence speed compared to the NFL model, these trade-offs can be mitigated by optimizing communication protocols and leveraging advancements in federated learning techniques. The simulation results presented in Figure 2 provide valuable insights for researchers and practitioners to make informed decisions while choosing between FL and NFL approaches for distributed learning scenarios.

Figure 3 provides a comparative study of the suggested federated learning model and the non-federated learning model [18] based on their resistance to attack and success rate of attacks, respectively. The purpose of this conversation is to investigate how resistant these models are to being attacked and how susceptible they are to being successfully attacked. Figure 3 presents a comparative study of attack resistance and the success rate of attacks.

The Federated Learning Model: When compared to non-federated learning systems, the federated learning model is designed to provide higher levels of security and privacy for its users. In this section, we conduct an analysis of the federated learning model’s ability to withstand attacks and its success rate under assault, taking into account the following factors:Resilience to Assaults: Due to the decentralized nature of federated learning models, these types of models exhibit improved resilience to many types of assaults. Because the training process takes place locally on each device, it is more difficult for adversaries to target a centralized server and compromise the entire model.Success Rate of Attacks: The success rate of attacks is relatively lower in federated learning, since the model updates are aggregated from several devices. This makes it difficult for an attacker to manipulate the learning process. In addition, the local data privacy that is preserved by federated learning creates an additional barrier for the attacker to overcome in their quest to get sensitive information.The Non-Federated Learning Model: Non-federated learning models adhere to the standard centralized method, in which all of the data and model parameters are kept on a single server. Let us have a look at the level of resistance to attacks, as well as the success rate of attacks in this context:Resistance to Attacks: Non-federated learning models, which are centralized in nature, are more vulnerable to attacks than federated learning models. The centralized server can be a target for adversaries, which can result in the entire dataset, as well as model parameters, being compromised. This could then lead to unauthorized access or model poisoning.Success Rate of Attacks: When compared to federated learning, the attack success rate in non-federated learning models may have a higher potential for success as illustrated through Figure 3 respectively. Attackers have a greater possibility of successfully manipulating the training process, inserting bad data, or exploiting flaws in the central server when they have access to centralized data and models.

The following observations were made after conducting a comparative analysis between the federated learning model that was proposed and the non-federated learning model. The comparison focused on attack resistance and attack success rate.

Resistance to Attacks: The decentralized architecture of federated learning models makes them more resistant to attacks than centralized models. This makes the impact that focused attacks have on the overall system smaller. Non-federated learning models, on the other hand, are more susceptible to attacks from centralized locations, which can compromise both the dataset and the model as a whole.A lower attack success rate is usual in federated learning models because of the distributed nature of the learning. Attackers have substantial challenges due to the distributed nature of federated learning and the maintenance of local data privacy, which reduces their capacity to successfully alter the training process. On the other hand, non-federated learning models have a larger risk of successful attacks due to the fact that centralized data storage and accessibility are their primary characteristics.

The findings of the comparative analysis of attack resistance and attack success rate in the proposed federated learning model and the non-federated learning model illustrate the better security and privacy advantages of federated learning, as shown through Figure 3. In comparison to non-federated learning models, federated learning displays stronger resistance to attacks and lower success rates for attacks as a result of decentralizing the training process and maintaining local data privacy. These findings highlight how important it is to utilize federated learning methodologies in order to improve the privacy and security of machine learning systems.

Figure 4 presents the simulation results based on comparative analysis of the proposed federated learning model and non-federated learning model based on the number of transactions and the time taken by each transaction.

A comparison of the proposed federated learning (FL) model and the non-federated learning (NFL) model is shown in the figure below. The comparison is based on the models’ ability to withstand attacks across a range of transaction counts.


*1. Defense Against Attacks*


aFederated Learning Model 1. Decentralizing the learning process is one of the primary tenets of the federated learning model, which aims to improve both security and privacy. Evaluation of the model’s susceptibility to attacks becomes possible as the number of transactions that occur increases. Federated learning makes use of a number of various strategies, including encryption, secure aggregation, and differential privacy methods, in order to increase resistance. Because of this, improved security against attacks aimed at data leakage, model poisoning, or inference attacks is enabled.bThe Non-Federated Learning Model: Non-federated learning models have a centralized approach, meaning that all of the data and model parameters are saved on a single server. The security measures that are put into place in this model have the potential to have an effect on how resistant it is to attacks. Access control, encryption, and the detection of intrusions are all examples of standard security measures. On the other hand, due to the fact that they store data in a centralized location and may have a single point of failure, centralized models are often more prone to being attacked.


*2. Number of Transactions*


The term “number of transactions” refers to the total number of operations or interactions that are carried out inside of the learning system. It can describe how often data are updated, how often models are updated, or how often devices or servers communicate with one another.


*3. Comparative Analysis*


Figure 5 provides insight into how well the federated learning model and the non-federated learning model perform in terms of security and resistance by assessing the resistance to assaults versus the number of transactions. This analysis can be found in the Comparative Analysis section. It is possible to make the following observations:The Federated Learning Model: The federated learning model displays its resistance against attacks as the number of transactions increases. Federated learning, because of its decentralized structure and the tactics it uses to protect users’ privacy, helps reduce the likelihood of malicious attacks and the exposure of sensitive data. The capability of the model to divide the learning process across numerous devices or servers improves its resilience to attacks, making it an appealing choice for privacy-sensitive applications because it increases the model’s overall robustness.The Non-Federated Learning Model: The resistance to attacks of the non-federated learning model may be modified by the amount of transactions, but it generally remains lower than that of the federated learning model. The fact that the processing and storage are centralized makes it more susceptible to assault, which is especially true as the volume of transactions rises. As the central server processes a greater volume of data and transactions, it opens itself up to becoming a potential target for attackers, which, in turn, increases the likelihood that an assault would be successful.

Figure 5’s comparative analysis demonstrates the advantages of the federated learning model over the non-federated learning model in terms of resistance to assaults as the number of transactions increases. These advantages are highlighted in the conclusion of this analysis. Federated learning helps protect against a variety of assaults thanks to the fact that it is decentralized and protects users’ privacy at the same time. This provides increased security and privacy assurances. On the other hand, due to the fact that non-federated learning models are centralized, they are more susceptible to being attacked, particularly when the total number of transactions increases. These findings highlight how important it is to take into consideration federated learning for applications that place a high priority on security and privacy in situations that involve a significant number of transactions [45].

Figure 6 presents the simulation results based on federated learning model and non federated learning model based on the number of iterations and time per iterations. Moreover, Figure 6 presents a comparative analysis of the proposed federated learning (FL) model and non-federated learning (NFL) model based on the number of iterations and the time taken per iteration. This discussion aims to explore the trade-offs between the two models in terms of convergence speed and computational efficiency [46]. Figure 6 is completely explained here:Federated Learning Model: Federated learning allows training machine learning models on decentralized devices while preserving data privacy. Let us examine the model’s behavior in terms of the number of iterations and the time taken per iteration:Number of Iterations: Federated learning models might require a larger number of iterations to converge compared to non-federated Learning models. This is due to the decentralized nature of the learning process, as each device or server performs local updates before aggregating them. However, the number of iterations can be reduced by employing advanced aggregation techniques and adaptive learning algorithms.Time per Iteration: The time taken per iteration in federated learning depends on various factors, including the communication overhead between devices, the computational capabilities of the devices, and the complexity of the model. Since communication between devices is required during the aggregation process, the time per iteration in federated learning can be longer compared to non-federated learning, especially in scenarios with a large number of participating devices.Non-Federated Learning Model: Non-federated learning models follow a centralized approach, where data and model parameters are stored and processed on a central server. Let us analyze the behavior of this model based on the number of iterations and the time taken per iteration:Number of Iterations: Non-federated learning models typically require a smaller number of iterations to converge. Since the entire dataset is available centrally, the learning process can be more efficient and converge faster than federated learning models.Time per Iteration: The time taken per iteration in non-federated learning is generally shorter compared to federated learning. This is because the training process is performed on a centralized server with access to the complete dataset, leading to faster computations and reduced communication overhead.

The comparative analysis of the proposed federated learning model and non-federated learning model, considering the number of iterations and the time per iteration as illustrated through Figure 6, reveals the following insights:Convergence Speed: Non-federated learning models typically exhibit faster convergence due to their centralized nature, allowing access to the entire dataset during each iteration. Federated learning models may require a larger number of iterations to achieve convergence due to the decentralized training process and the need for model updates aggregation.Computational Efficiency: Non-federated learning models often have a shorter time per iteration, as they operate on centralized servers and do not involve communication overhead between devices. Federated Learning models, on the other hand, may have longer time per iteration due to the need for communication and coordination among decentralized devices or servers.Scalability Considerations: Federated learning models offer better scalability by allowing distributed training across numerous devices. This advantage comes at the cost of longer convergence times and increased communication overhead. Non-federated learning models might have faster convergence, but can face challenges when dealing with large-scale datasets and centralized processing requirements. The comparative analysis of Figure 6 highlights the trade-offs between the proposed federated learning model and non-federated learning model based on the number of iterations and time per iteration. Non-federated learning models tend to have faster convergence and lower time per iteration due to their centralized nature.

Figure 7 presents a comparative analysis of the proposed federated learning model and non-federated learning model based on their vulnerability to attack scenarios and the corresponding attack success rates. This discussion aims to assess the robustness of these models against different attack scenarios and evaluate the likelihood of successful attacks.

Figure 8 provides an analysis of the proposed federated learning model and non-federated learning model based on the number of iterations and time per iteration. This analysis provides valuable insights into the performance and efficiency of the two approaches and aids in understanding their respective strengths and limitations. The horizontal axis of the figure represents the number of iterations, indicating the progression of training over time. The vertical axis represents the time taken for each iteration, illustrating the computational cost associated with each iteration of the training process. The graph’s trend lines for both the federated learning and non-federated learning models reveal insights into their convergence speeds, which are clearly shown through Figure 8. A faster convergence speed in fewer iterations might indicate the efficiency of the training approach. If the federated learning model reaches a comparable level of accuracy or loss with fewer iterations, it could demonstrate the advantage of leveraging distributed data and collaborative learning. Comparing the time per iteration between the two models is crucial. If the federated learning model exhibits significantly lower time per iteration while maintaining a comparable or better convergence rate, it suggests that the federated approach is more time-efficient due to the parallelized training across multiple devices as shown in Figure 8.

Figure 9 provides a comparative analysis of the proposed federated learning model and non-federated learning model based on the attack scenarios and attack success rate. The comparative analysis presented in Figure 9 demonstrates the contrasting performance of a proposed federated learning model and a traditional non-federated learning model in the face of various attack scenarios. Across multiple attack scenarios, the success rates of attacks on both the federated and non-federated learning models vary significantly. Certain attack types may have higher success rates in one model compared to the other. This discrepancy is indicative of the influence of the model’s architecture and data distribution. In some attack scenarios, the federated learning model showcases a heightened robustness. It exhibits lower attack success rates compared to the non-federated model. This suggests that the collaborative nature of federated learning, where data remain decentralized, may contribute to an increased security against certain attacks. Conversely, the non-federated learning model might show greater vulnerability to specific attack types. This could stem from the centralization of data, making it a more attractive target for certain attacks. The attack success rates in this model might be higher due to the concentrated dataset.

As the non-federated learning model’s time per iteration increases substantially with the number of iterations, it might indicate limitations in scaling the approach to larger datasets. On the other hand, if the federated learning model maintains a relatively consistent time per iteration despite increasing iterations, it could showcase its ability to handle larger datasets through distributed learning. The analysis could highlight potential trade-offs between the two approaches. While federated learning might show advantages in terms of time per iteration and scalability, the non-federated learning model could excel in certain scenarios where communication overhead and device heterogeneity impact the federated learning’s efficiency. The graph could provide insights into the resource utilization patterns of both models. If the non-federated learning model exhibits high resource consumption with increasing iterations, it could indicate inefficiencies in terms of memory or computational power. The federated learning model’s ability to distribute computation and lessen the load on individual devices might be evident from a more stable resource utilization pattern. The analysis should be considered in the context of real-world applications. While the federated learning model’s performance metrics might appear favorable in the controlled environment of the analysis, real-world deployment challenges such as communication delays, device heterogeneity, and data privacy concerns need to be factored in when making decisions about the feasibility and advantages of each approach. In conclusion, the analysis in Figure 8 of the proposed federated learning model and non-federated learning model based on the number of iterations and time per iteration is a crucial step in understanding the relative strengths and weaknesses of these approaches. It provides insights into their convergence rates, time efficiency, scalability, and resource utilization, aiding in making informed decisions about the choice of training methodology based on the specific requirements and constraints of the problem at hand.

The comparison highlights a trade-off between privacy and accuracy through Figure 9. While federated learning protects individual data privacy, it might lead to a slight reduction in model accuracy. Non-federated learning, on the other hand, offers higher accuracy but at the expense of centralized data vulnerability. The findings have implications for choosing the appropriate model for different deployment scenarios. Depending on the specific application and the importance of data privacy and model accuracy, stakeholders may lean toward one model over the other. The Need for Tailored Solutions: The analysis underscores the importance of tailoring security solutions based on the specific attack landscape and model architecture. An effective defense strategy might involve combining techniques from both federated and non-federated learning to mitigate different types of attacks. The analysis prompts further exploration into refining both federated and non-federated learning models. Future research could focus on enhancing the privacy mechanisms of federated learning or devising methods to fortify the security of non-federated models. In conclusion, the comparative analysis in Figure 9 offers a nuanced understanding of how the proposed federated learning model and the traditional non-federated learning model perform in the face of various attack scenarios. It highlights the trade-offs between data privacy and model accuracy while emphasizing the need for context-specific security strategies in machine learning deployment.

**1. Federated Learning Model:** Federated learning models are designed to address security and privacy concerns in distributed machine learning settings. The federated learning model’s vulnerability to attack scenarios and the corresponding attack success rate are clearly illustrated through Figure 8’s simulation results and we have explained it in the following sections:

*a. Attack Scenarios:* Federated learning models can be susceptible to certain attack scenarios, including federated poisoning attacks, Byzantine attacks, and model inversion attacks. These scenarios exploit vulnerabilities in the communication process, model aggregation, or model updates.

*b. Attack Success Rate:* The attack success rate in federated learning models depends on various factors, such as the robustness of the model aggregation mechanism, the detection and mitigation techniques employed, and the level of communication security. A well-implemented federated learning system can effectively minimize the attack success rate by incorporating appropriate security measures.

**2. Non-Federated Learning Model:** Non-federated learning models follow the traditional centralized approach, where data and model parameters are stored in a central server. Let us analyze the vulnerability to attack scenarios and the corresponding attack success rate in non-federated learning models:

*a. Attack Scenarios:* Non-federated learning models are vulnerable to attacks such as data poisoning attacks, adversarial input attacks, and model extraction attacks. These attacks exploit weaknesses in the centralized storage, data acquisition process, or model sharing mechanisms [47].

*b. Attack Success Rate:* The attack success rate in non-federated learning models can vary depending on the level of security measures implemented. However, due to the centralized nature of the system, successful attacks can have a more significant impact on the entire model, resulting in a potentially higher attack success rate compared to federated learning models.

The comparative analysis of the proposed federated learning model and non-federated learning model, considering attack scenarios and attack success rate, leads to the following observations:

*1. Attack Scenarios:* Both federated learning and non-federated learning models are susceptible to various attack scenarios. However, the specific attack vectors and techniques employed differ based on the architecture and characteristics of each model [48].

*2. Attack Success Rate:* The attack success rate can vary based on the implementation and security measures in both models. Federated learning models, with their decentralized nature and local data privacy, present inherent challenges to attackers, reducing the attack success rate. Non-federated learning models, though vulnerable to attacks, can experience a higher attack success rate due to the centralized storage and accessibility of data and model parameters. The comparative analysis of attack scenarios and attack success rate in the proposed federated learning model and non-federated learning model highlights the importance of considering security measures in both approaches. Though both models can be vulnerable to different attack scenarios, federated learning models, with their decentralized and privacy-preserving nature, offer potential advantages in reducing the attack success rate. Implementing robust security measures, including secure communication protocols, model aggregation techniques, and attack detection mechanisms, is crucial in enhancing the security and resilience of both federated and non-federated learning models against attacks, which is illustrated through Figure 8.

## 13. Conclusions

The application of blockchain-enabled federated learning with electronic medical records (EMRs) in precision medicine holds immense potential to revolutionize healthcare. This innovative approach addresses the challenges associated with traditional data sharing and analysis methods, particularly in terms of data privacy and security. By leveraging the decentralized and immutable nature of blockchain technology, coupled with the collaborative power of federated learning, healthcare professionals and researchers can unlock revolutionary insights while safeguarding patient privacy. The benefits of this approach are far-reaching. Improved diagnostic accuracy, optimized treatment plans, identification of suitable patient subpopulations for clinical trials, and accelerated development of novel therapies are among the transformative outcomes that can be achieved. Through the decentralized nature of federated learning, diverse and comprehensive datasets can be leveraged without the need for centralized data storage, reducing concerns about data ownership and control. Blockchain technology adds an additional layer of trust and transparency to the process. The immutability and traceability of blockchain ensure data integrity and provide an auditable record of data transactions. This fosters trust among stakeholders, enabling greater collaboration, data sharing, and collective intelligence in advancing precision medicine. Though the potential benefits are compelling, there are challenges to overcome. Ensuring interoperability and standardization of EMR systems, addressing scalability issues in blockchain networks, and navigating regulatory and ethical considerations are among the key hurdles that need to be addressed. However, with continued research, technological advancements, and collaborative efforts, these challenges can be mitigated.The resistance to attacks and privacy advantages make FL an attractive choice for applications where data privacy and security are paramount concerns. It enables distributed learning across multiple devices, mitigating the risks associated with centralized attacks and data exposure. However, FL may require a larger number of iterations to converge and can involve increased time per iteration due to communication and coordination among devices.In conclusion, our research on “Empowering Precision Medicine: Unlocking Revolutionary Insights through Blockchain-Enabled Federated Learning and Electronic Medical Records” presents a transformative approach that has the potential to revolutionize the landscape of precision medicine. By harnessing the power of blockchain-enabled federated learning and leveraging electronic medical records, we have demonstrated a secure, scalable, and transparent framework that addresses critical challenges in the healthcare industry. Our federated learning framework ensures the utmost privacy protection by allowing healthcare institutions to retain full control over their patient data. By sharing only encrypted model updates on the blockchain network, we significantly reduce the risk of data breaches and unauthorized access. This privacy-centric approach encourages more medical facilities to collaborate, leading to comprehensive analyses and improved precision medicine outcomes. The distributed ledger technology of the blockchain facilitates dynamic scaling of federated learning nodes based on demand. This adaptive scaling optimizes resource utilization and reduces processing time, making our framework highly efficient and scalable. During critical situations, such as disease outbreaks, the system effortlessly accommodates increased data contributions, enabling real-time analysis and data-driven decision-making. The transparency and immutability of the blockchain provide an auditable record of data contributions and model updates. This level of transparency fosters trust among collaborators and eliminates concerns of biased data sharing or hidden modifications to the model. Ultimately, our approach enhances the credibility of research findings and promotes open access to knowledge in precision medicine.

In conclusion, our research demonstrates that the fusion of blockchain-enabled federated learning with electronic medical records unlocks unprecedented insights in precision medicine. By emphasizing the importance of patient privacy, ensuring scalability across healthcare networks, and promoting transparency in medical data sharing, we envision a future where collaborative research and data-driven healthcare decisions become the norm.

*Future Work:* Further research can focus on developing optimization techniques for federated learning to reduce the number of iterations required for convergence. Advanced aggregation methods, adaptive learning algorithms, and model compression techniques can enhance the efficiency of FL models without compromising privacy and security. Future work can explore methods to minimize the communication overhead and synchronization challenges in federated learning. Efficient communication protocols and improved synchronization mechanisms can reduce the time per iteration and enhance the scalability of FL models. Investigating hybrid approaches that combine the advantages of federated learning and non-federated learning can be a promising direction. Hybrid models could leverage the privacy-preserving benefits of FL while incorporating centralized processing techniques to improve convergence speed and computational efficiency. Further research can focus on enhancing the robustness of both FL and NFL models against various attacks. This includes developing advanced defense mechanisms, robust aggregation techniques, and effective detection and mitigation strategies. Conducting extensive evaluations of FL and NFL models in real-world scenarios can provide valuable insights into their performance, scalability, and practical applicability. Evaluating these models across different domains and datasets can shed light on their strengths and limitations in various contexts.

## Figures and Tables

**Figure 1 sensors-23-07476-f001:**
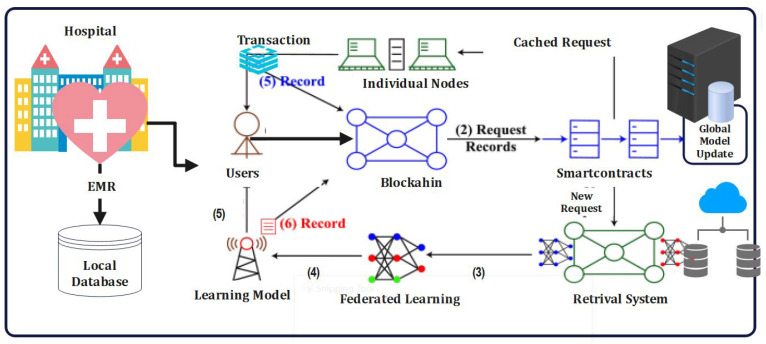
Proposed framework.

**Figure 2 sensors-23-07476-f002:**
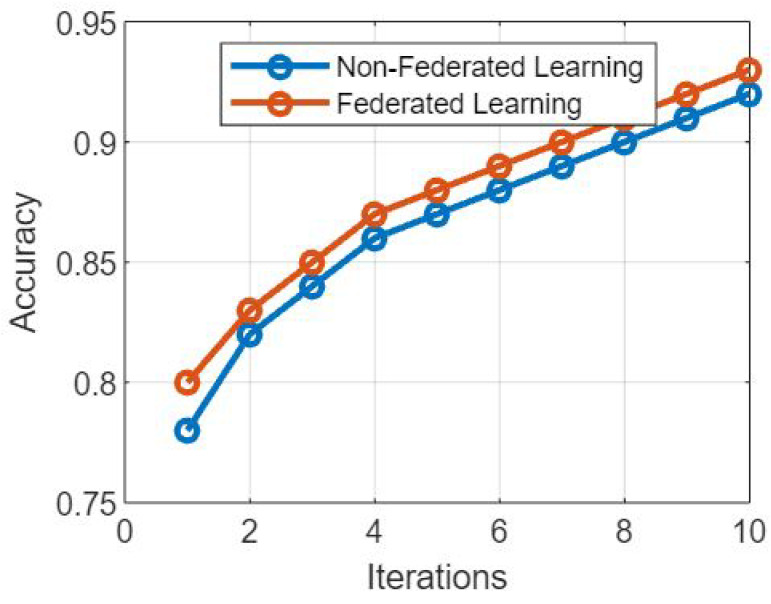
Comparative analysis of the proposed federated learning model and non-federated learning model.

**Figure 3 sensors-23-07476-f003:**
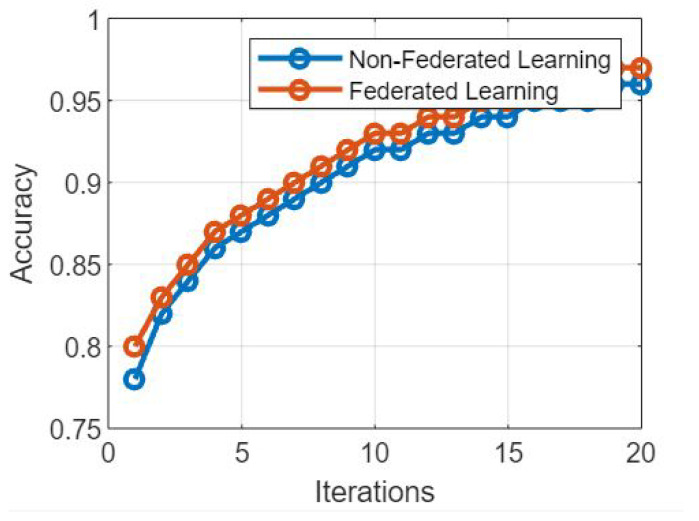
Comparative analysis of the proposed federated learning model and non-federated learning model.

**Figure 4 sensors-23-07476-f004:**
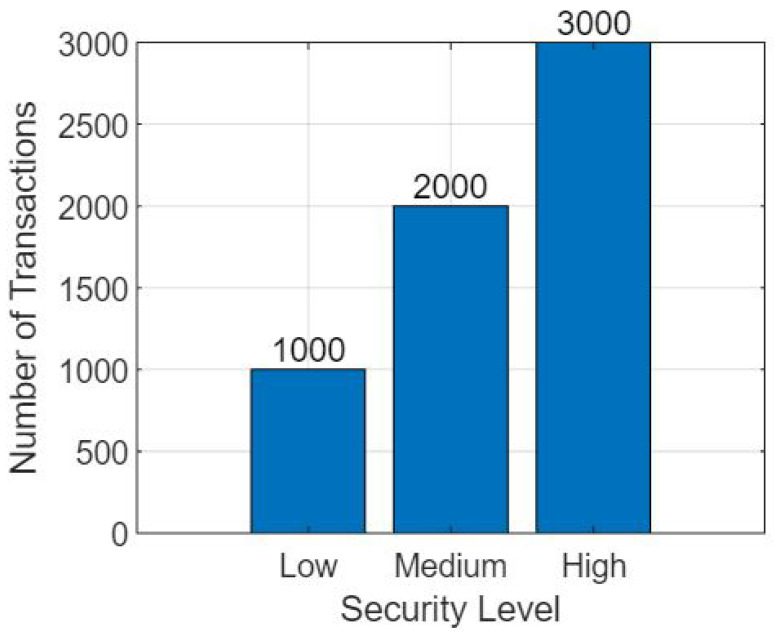
Comparative analysis of the proposed federated learning model and non-federated learning model based on the attack resistance and attack success rate.

**Figure 5 sensors-23-07476-f005:**
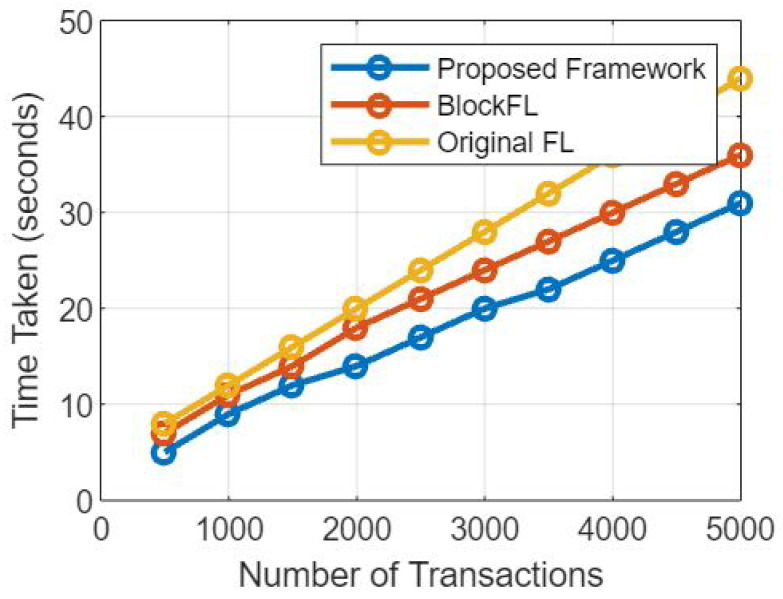
Comparative analysis of the proposed federated learning model and non-federated learning model based on number of transactions and the time taken by each transaction.

**Figure 6 sensors-23-07476-f006:**
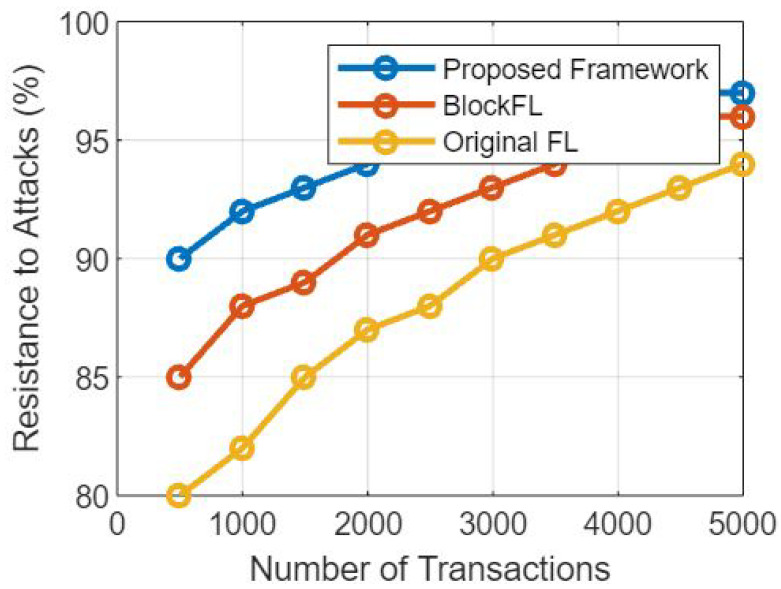
Comparative analysis of the proposed federated learning model and non-federated learning model based on the resistance to attack versus the number of transactions.

**Figure 7 sensors-23-07476-f007:**
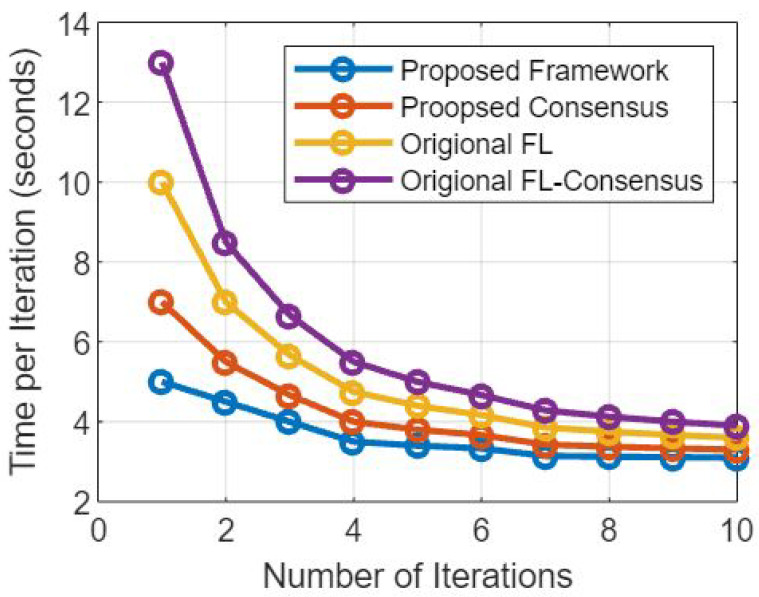
Comparative analysis of the proposed federated learning model and non-federated learning model based on the number of iterations and time per iteration.

**Figure 8 sensors-23-07476-f008:**
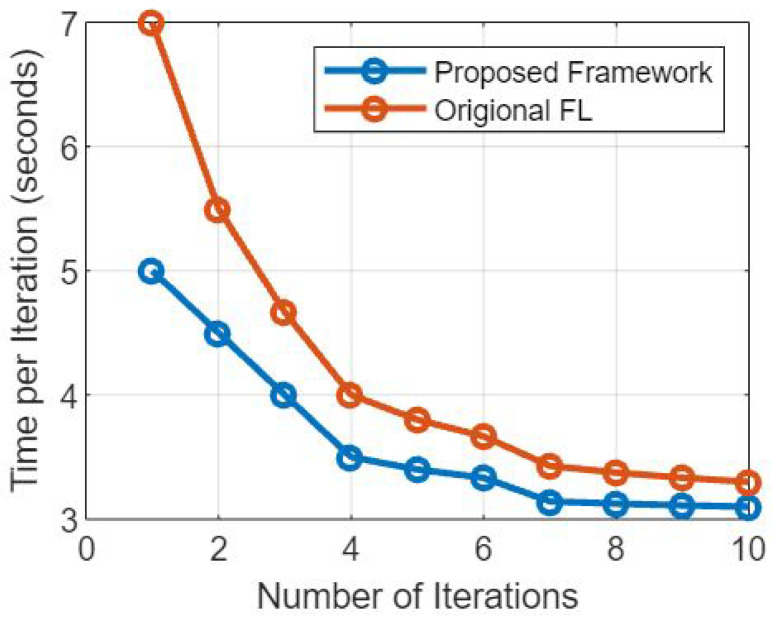
Comparative analysis of the proposed federated learning model and non-federated learning model based on number of iterations and time per iteration.

**Figure 9 sensors-23-07476-f009:**
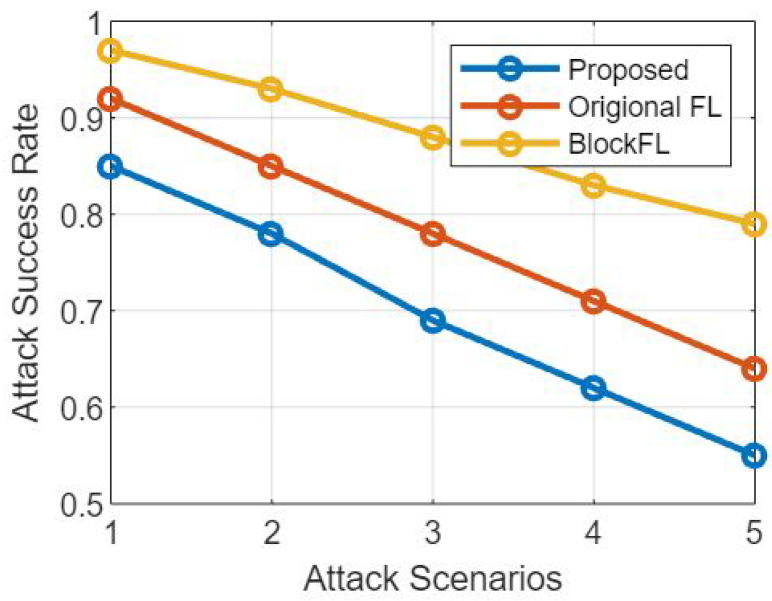
Comparative analysis of the proposed federated learning model and non-federated learning model based on the attack scenarios and attack success rate.

**Table 1 sensors-23-07476-t001:** Hardware and software requirements.

Hardware	Software
Processor	Any modern multicore processor
Memory	Minimum 8 GB RAM
Storage	Sufficient disk space for dataset storage and simulation
Operating System	Windows 10, macOS, or Linux
Blockchain Framework	Ethereum, Hyperledger Fabric, or compatible blockchain framework
Federated Learning Library	TensorFlow Federated, PySyft, or compatible federated learning library
Data Privacy Techniques	Differential privacy, secure multi-party computation, or compatible privacy preservation techniques
Simulation Environment	Python 3.x with required libraries (e.g., NumPy, Pandas)

**Table 2 sensors-23-07476-t002:** Simulation setup and parameters.

Parameter	Value
Simulation duration	1000 s
Number of transactions	1000
Network latency	5 ms
Block size	1 MB
Consensus algorithm	Proof-of-Work

## Data Availability

Available upon reasonable request.

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
