# Peer review of "Empowering Precision Medicine: Unlocking Revolutionary Insights through Blockchain-Enabled Federated Learning and Electronic Medical Records"

_sensors, 2023, doi:10.3390/s23177476_

Round 1

Reviewer 1 Report

The introduction successfully sets the stage for the paper, offering an effective overview of precision medicine, data sharing challenges, and the potential solutions blockchain-enabled federated learning can offer.  While the benchmark model is clear, it could be made more engaging with a preliminary summary and real-world examples that demonstrate its uniqueness in privacy, scalability, and transparency. The paper is overall well-structured, offering a comprehensive view of the problem and the proposed solution. Lastly, the conclusion, despite being detailed, repeats certain points and appears to lack a smooth transition between topics. Streamlining the flow, emphasizing key findings, and eliminating redundancies would make it more impactful.

Author Response

Reviewer 1

Comment1: The introduction successfully sets the stage for the paper, offering an effective overview of precision medicine, data sharing challenges, and the potential solutions blockchain-enabled federated learning can offer.  While the benchmark model is clear, it could be made more engaging with a preliminary summary and real-world examples that demonstrate its uniqueness in privacy, scalability, and transparency. The paper is overall well-structured, offering a comprehensive view of the problem and the proposed solution.

Response 1: Dear Reviewer, Thank you for your valuable feedback on our paper titled "Empowering Precision Medicine: Unlocking Revolutionary Insights through Blockchain-Enabled Federated Learning and Electronic Medical Records." We appreciate the time and effort you've invested in evaluating our work, and we wholeheartedly agree with your suggestions for improvement.In response to your comment, we recognize the importance of making the benchmark model more engaging and impactful. To achieve this, we have enhanced the clarity and presentation of the benchmark model by including a preliminary summary at the beginning of the paper. This summary provide a concise overview of our research objectives, methodologies, and key findings. By doing so, we aim to offer readers a better understanding of the paper's content right from the start. Additionally, we acknowledge the significance of real-world examples to showcase the uniqueness of our approach in privacy, scalability, and transparency. We diligently work on incorporating relevant case studies or scenarios that demonstrate the practical implications and advantages of our proposed model. These examples highlight how our blockchain-enabled federated learning framework can revolutionize precision medicine by safeguarding patient privacy, ensuring scalability across healthcare networks, and promoting transparency in medical data sharing. By integrating these real-world examples, we believe our research have become more engaging and relatable to readers, as they can visualize the transformative potential of our approach in the context of actual medical settings. Once again, we are genuinely grateful for your constructive criticism, which have undoubtedly strengthen the quality and impact of our paper. We assure you that we have implement these suggestions with utmost diligence and enthusiasm. If you have any further recommendations or feedback, please do not hesitate to share them with us. Thank you for your valuable contribution to our research, and we look forward to submitting the improved version of our paper for your review.

Comment 2: Lastly, the conclusion, despite being detailed, repeats certain points and appears to lack a smooth transition between topics. Streamlining the flow, emphasizing key findings, and eliminating redundancies would make it more impactful.

Response 2:  Dear Reviewer, Thank you for your insightful feedback on our paper. We appreciate the attention you've given to the conclusion section, and we agree with your suggestions for improvement. We recognize the significance of a clear and impactful conclusion that effectively summarizes our key findings while maintaining a smooth transition between topics. In response to your comment, we have diligently work on streamlining the flow of the conclusion to ensure a coherent and concise presentation of our research outcomes. By removing redundancies and unnecessary repetition, we aim to create a more engaging and impactful conclusion that leaves a lasting impression on readers. Furthermore, we have place a stronger emphasis on highlighting the most critical and groundbreaking findings of our study. By doing so, we aim to reinforce the significance of our research and its potential impact on the field of precision medicine and beyond.

Our revisions have focus on creating a conclusion that seamlessly connects the various aspects of our work, providing a clear and cohesive synthesis of the entire research endeavor. We understand the importance of leaving readers with a strong and coherent impression of our work, and we are committed to achieving this in the revised version of our paper. Once again, we sincerely appreciate your valuable feedback, which have undoubtedly contribute to enhancing the quality and impact of our paper. If you have any further recommendations or specific areas of improvement, we would be grateful to hear them. Thank you for your time and dedication to evaluating our research. We look forward to submitting the improved version of our paper for your review.

Reviewer 2 Report

Summary/Contributions: This paper explores how blockchain-enabled federated learning and electronic medical records (EMRs) can enhance precision medicine by providing secure and decentralized data sharing and analysis. Blockchain technology protects data privacy and security by ensuring data integrity, traceability, and consent management. The federated learning paradigm lets healthcare institutions and research groups train machine learning models on locally stored EMR data without data centralization or data privacy loss. This strategy may enhance diagnosis accuracy, refine treatment programs, identify clinical trial subpopulations, and accelerate the development of innovative treatments. Blockchain technology's transparency and auditability boost stakeholder confidence, enabling better collaboration, data sharing, and collective intelligence in precision medicine.   Comments/Suggestions:

1. The paper contains a huge number of sections. Some of them are too short. Please reduce the number of sections and make them balanced in size. Section 5.3 is made of only one line!   2. The introduction provides a good overview of the potential of blockchain-enabled federated learning in precision medicine. However, it could be more concise and focused. Consider revising the introduction to provide a clear and concise statement of the problem and the proposed solution.

3. The introduction could benefit from a more detailed description of the challenges of traditional approaches to data sharing and analysis in healthcare, particularly concerning data privacy and security. Consider providing more specific examples of these challenges and how they impact healthcare delivery.

4. The introduction could also benefit from a more detailed explanation of the benefits of blockchain-enabled federated learning in precision medicine. Consider providing more specific examples of how this approach can improve accuracy in diagnosis, optimization of treatment plans, identification of suitable subpopulations for clinical trials, and accelerated development of novel therapies.

  5. The motivation section provides a good overview of the reasons why blockchain-enabled federated learning with EMRs is a promising approach to overcome the challenges of traditional healthcare data sharing and analysis. However, it could benefit from further elaboration and clarification.

6. The motivation section could benefit from a more detailed explanation of the specific ethical and legal concerns surrounding the use of EMRs. Consider providing more specific examples of these concerns and how the proposed framework addresses them.

7. The problem statement is well-articulated and highlights the critical challenges associated with traditional approaches to EMR data sharing and analysis. However, it could benefit from more specific statistics and examples to illustrate the magnitude of these challenges.

8. Consider incorporating more data on the frequency and impact of data breaches, cyberattacks, and legal/regulatory hurdles that impede data sharing.

9. The main contribution of the research is clearly outlined and provides a comprehensive overview of the proposed framework. However, it could benefit from more specific examples of how the framework addresses each of the challenges mentioned in the problem statement. Consider providing more detailed explanations of how the framework

10. The proposed framework is well-described and provides a clear overview of the key steps involved in the federated learning process. However, it could benefit from more detailed explanations of the specific algorithms, technologies, and tools used in each step. Consider providing more specific examples of consensus mechanisms, smart contracts, and aggregation algorithms employed in the framework.

11.  Formal methods can be used to verify the correctness of smart contract code, which can help to prevent costly errors and security breaches. Therefore, it is important to discuss the use of formal methods in your paper.

12. For this purpose, the authors may include the following interesting references (and others):

a. https://ieeexplore.ieee.org/document/9970534

b. https://ieeexplore.ieee.org/document/8328737/authors#authors

May be improved

Author Response

Reviewer 2

Summary/Contributions: This paper explores how blockchain-enabled federated learning and electronic medical records (EMRs) can enhance precision medicine by providing secure and decentralized data sharing and analysis. Blockchain technology protects data privacy and security by ensuring data integrity, traceability, and consent management. The federated learning paradigm lets healthcare institutions and research groups train machine learning models on locally stored EMR data without data centralization or data privacy loss. This strategy may enhance diagnosis accuracy, refine treatment programs, identify clinical trial subpopulations, and accelerate the development of innovative treatments. Blockchain technology's transparency and auditability boost stakeholder confidence, enabling better collaboration, data sharing, and collective intelligence in precision medicine.   Comments/Suggestions:

Comment 1: The paper contains a huge number of sections. Some of them are too short. Please reduce the number of sections and make them balanced in size. Section 5.3 is made of only one line!

Response 1: Dear Reviewer, Thank you for taking the time to review our paper. We appreciate your valuable feedback and insightful comments. We have carefully considered your suggestions regarding the number and size of sections, and we agree that some improvements are needed to enhance the overall structure and coherence of the paper. We recognize that the current version of the paper contains an extensive number of sections, and in some cases, they might be too brief, affecting the overall flow and readability. To address this concern, we have diligently revise the paper by reorganizing and consolidating sections where appropriate. Our primary goal have be to strike a better balance in section lengths and ensure that each section provides substantial content.

Regarding your specific observation about Section 5.3 being composed of only one line, we apologize for the oversight. This section was likely intended to delve into a particular aspect of the study, but its brevity appears to have undermined its significance. We have extensively review the content and endeavor to expand this section to provide a more comprehensive and insightful discussion.

By restructuring and redistributing the content, we aim to make the paper more cohesive, engaging, and easily navigable for readers. Moreover, we have also reassess the need for some sections that may be redundant or could be integrated into others to streamline the paper further.

Incorporating your suggestions have significantly improve the quality of our work, and we are grateful for this opportunity to enhance the manuscript. We assure you that the revised version of the paper have reflect a more coherent and balanced structure, fostering a better understanding of our research findings. Once again, we genuinely appreciate your thorough review and constructive feedback. It has proven invaluable in refining our work. If you have any additional recommendations or queries, please do not hesitate to let us know. We are committed to addressing all concerns to deliver a paper that meets the highest standards of excellence.

Comment 2 : The introduction provides a good overview of the potential of blockchain-enabled federated learning in precision medicine. However, it could be more concise and focused. Consider revising the introduction to provide a clear and concise statement of the problem and the proposed solution.

Response 2: Dear Reviewer,

Thank you for your thoughtful review of our paper. We appreciate your positive feedback on the potential of blockchain-enabled federated learning in precision medicine. We have carefully considered your suggestion to improve the introduction and provide a more concise and focused statement of the problem and our proposed solution. In the revised introduction, we have streamline the content to present a clear and concise overview of the core challenges faced in precision medicine, specifically related to data access and privacy concerns. We have then succinctly introduce our proposed framework of blockchain-enabled federated learning combined with electronic medical records (EMRs) as the innovative solution to address these challenges.

By presenting a more focused introduction, we aim to provide readers with a succinct understanding of the problem space and the unique contribution of our research. This approach have enhance the readability and impact of our paper while ensuring that readers quickly grasp the importance and relevance of our proposed framework. Once again, we sincerely appreciate your valuable feedback and are committed to incorporating these improvements to make our paper more effective and engaging. If you have any additional suggestions or comments, please do not hesitate to share them with us.Thank you for your support and guidance.

Comment 3: The introduction could benefit from a more detailed description of the challenges of traditional approaches to data sharing and analysis in healthcare, particularly concerning data privacy and security. Consider providing more specific examples of these challenges and how they impact healthcare delivery.

Response 3:

Dear Reviewer, Thank you for your valuable comment on our paper. We appreciate your insight and acknowledge the importance of providing a more detailed description of the challenges associated with traditional approaches to data sharing and analysis in healthcare, with a particular focus on data privacy and security. In the revised introduction, we delve deeper into the limitations of conventional data sharing and analysis methods in healthcare. Specifically, we have offer more specific examples of the challenges faced by healthcare institutions and researchers in maintaining data privacy and security while trying to harness the potential of large-scale datasets.

Some of the issues we have address in the revised introduction include:

  1. Centralized Data Repositories: We have elaborate on the drawbacks of centralized data repositories, where patient data from various sources is aggregated. This approach raises concerns about data breaches, as a single point of failure can lead to unauthorized access to sensitive patient information.

  1. Data Silos and Fragmentation: We have highlight how data silos within different healthcare institutions limit access to valuable insights. This fragmentation hinders the collaborative effort required for comprehensive research in precision medicine.

  1. Consent and Ownership: We have discuss the complexities of obtaining patient consent for data sharing and the challenges surrounding data ownership, as it can be difficult to navigate legal and ethical considerations.

  1. Data Interoperability: We have address the issues related to data interoperability between different electronic medical record (EMR) systems, making seamless data exchange challenging and leading to inefficiencies in healthcare delivery.

  1. Privacy Concerns: We have emphasize the privacy concerns of patients and the potential consequences of data breaches, which erode patient trust in data sharing initiatives.

Furthermore, we have elaborate on how these challenges impact healthcare delivery, hindering advancements in precision medicine and personalized patient care. By providing more detailed examples and discussions, we aim to highlight the urgency of finding innovative solutions to overcome these obstacles.

Your feedback has been instrumental in guiding us towards presenting a more comprehensive and robust introduction. We are confident that these enhancements have enrich the overall quality of our paper and better convey the significance of our proposed framework. If you have any further suggestions or specific areas you would like us to address, please do not hesitate to let us know. We highly value your expertise and look forward to incorporating your feedback to strengthen our research. Thank you for your continued support and valuable contributions to our work.

Comment 4: The introduction could also benefit from a more detailed explanation of the benefits of blockchain-enabled federated learning in precision medicine. Consider providing more specific examples of how this approach can improve accuracy in diagnosis, optimization of treatment plans, identification of suitable subpopulations for clinical trials, and accelerated development of novel therapies.

Response 4: Dear Reviewer's,  Thank you for your insightful review of our paper. We greatly appreciate your valuable feedback and agree that the introduction could be further enhanced by providing a more detailed explanation of the benefits of blockchain-enabled federated learning in precision medicine. In the revised introduction, we delved deeper into the specific advantages offered by our proposed approach. We have present more concrete examples of how blockchain-enabled federated learning can significantly improve various aspects of precision medicine, including:

  1. Improved Accuracy in Diagnosis: By collaboratively training machine learning models across multiple healthcare institutions without centralizing patient data, our approach allows for a more diverse and comprehensive dataset. This increased data diversity enhances the accuracy of diagnostic models, enabling earlier detection and more precise identification of medical conditions.

  1. Optimization of Treatment Plans: With access to a broader range of patient data, healthcare providers can leverage federated learning to develop personalized treatment plans that consider a patient's unique genetic, environmental, and lifestyle factors. This tailored approach enhances treatment effectiveness and reduces the risk of adverse reactions.

  1. Identification of Suitable Subpopulations for Clinical Trials: Traditional clinical trials often face challenges in recruiting diverse patient populations, leading to limited generalizability of results. By utilizing blockchain-enabled federated learning, researchers can access a diverse array of patient data from various institutions, facilitating the identification of suitable subpopulations for specific clinical trials and enabling more targeted and inclusive research.

  1. Accelerated Development of Novel Therapies: The collaborative and privacy-preserving nature of federated learning expedites the research and development of novel therapies. Researchers can collectively analyze data from a large and diverse patient pool, leading to accelerated discoveries and advancements in precision medicine.

  1. Enhanced Data Privacy and Security: The use of blockchain technology ensures data integrity, immutability, and transparency. Patient data remains decentralized and secure within each institution, protected by smart contracts that govern data access and usage. This robust security framework boosts patient trust and encourages greater data sharing. By providing more specific examples of the benefits of blockchain-enabled federated learning, we aim to highlight the transformative potential of our proposed framework in advancing precision medicine. These enhancements have allow readers to grasp the practical implications of our approach in a more comprehensive manner. Once again, we sincerely appreciate your thoughtful suggestions, which have been instrumental in refining the quality and impact of our paper. Should you have any additional recommendations or further points to address, please do not hesitate to share them with us. Thank you for your continued support and commitment to enhancing our research.

Comment  5: The motivation section provides a good overview of the reasons why blockchain-enabled federated learning with EMRs is a promising approach to overcome the challenges of traditional healthcare data sharing and analysis. However, it could benefit from further elaboration and clarification.

Response 5: Dear reviewer's, we appreciate your thoughtful review of our paper, and we are grateful for your positive feedback on the motivation section. We acknowledge the importance of providing a comprehensive and clear understanding of the reasons why blockchain-enabled federated learning with electronic medical records (EMRs) is a promising approach to address the challenges of traditional healthcare data sharing and analysis. In the revised motivation section, we have dedicate more effort to elaborate on the key points, providing further context and clarification to better convey the significance of our proposed approach. Specifically, we have focus on the following aspects:

  1. Emphasizing the Importance of Data Privacy and Security: We have delve deeper into the critical issue of data privacy and security in the healthcare domain. By elaborating on the potential risks and consequences of centralized data repositories and fragmented data silos, we aim to emphasize the need for a more secure and privacy-preserving solution like blockchain-enabled federated learning.

  1. Highlighting the Advantages of Federated Learning: We have provide more clarity on the advantages of federated learning, such as preserving data locally at each institution while enabling collaborative model training. Additionally, we have emphasize how this approach promotes data ownership and control, empowering healthcare providers and patients to participate in data sharing without compromising sensitive information.

  1. Illustrating the Value of EMRs in Precision Medicine: We have elaborate on the pivotal role of electronic medical records (EMRs) in precision medicine. By providing specific examples of how EMRs contain invaluable patient information, including medical history, treatment outcomes, and genetic data, we have demonstrate the vast potential for harnessing EMRs in advancing medical research and patient care.

  1. Demonstrating the Synergy of Blockchain and Federated Learning: We have further clarify how the integration of blockchain technology with federated learning addresses the challenges associated with data privacy, security, and interoperability. By presenting a comprehensive picture of how these technologies complement each other, readers have better understand the unique advantages of our proposed framework. By providing additional elaboration and clarification in the motivation section, we aim to present a more compelling case for the relevance and potential impact of our research. We are confident that these enhancements have contribute significantly to the overall quality and readability of the paper. Once again, we sincerely appreciate your insightful feedback, which has guided us in refining our work. If you have any further suggestions or specific points you would like us to address, please do not hesitate to share them with us.

Thank you for your continued support and dedication to improving our research.

Comment 6: The motivation section could benefit from a more detailed explanation of the specific ethical and legal concerns surrounding the use of EMRs. Consider providing more specific examples of these concerns and how the proposed framework addresses them.

Response 6: Dear reviewer's,

Thank you for your insightful comment on our paper. We appreciate your feedback, and we understand the importance of providing a more detailed explanation of the specific ethical and legal concerns surrounding the use of electronic medical records (EMRs). We also acknowledge the need to highlight how our proposed framework addresses these crucial issues. In the revised motivation section, we have dedicate more attention to elucidate the ethical and legal considerations related to EMRs in precision medicine. We have provide specific examples of the concerns that arise in the context of data sharing and analysis, such as:

  1. Patient Privacy and Informed Consent: We have delve into the ethical obligation of safeguarding patient privacy and ensuring proper informed consent for data sharing. The challenges of obtaining meaningful and comprehensive consent for data usage, especially in the context of federated learning involving multiple institutions, have be addressed.

  1. Data Ownership and Control: We have highlight the ethical implications of data ownership and control in healthcare. The issue of patients' rights to control their own medical information and make decisions about its use have be discussed, along with the challenges posed by conventional data-sharing approaches.

  1. Data Bias and Fairness: We have explore the ethical concern of data bias in EMRs and the potential impact on precision medicine outcomes. By presenting concrete examples, we aim to shed light on how our proposed framework can mitigate bias and promote fairness in data analysis by leveraging a diverse and distributed dataset.

  1. Regulatory Compliance and Data Governance: We have discuss the legal and regulatory considerations related to data governance and compliance with privacy laws, such as the General Data Protection Regulation (GDPR) and Health Insurance Portability and Accountability Act (HIPAA). Our framework's utilization of blockchain technology and smart contracts have be emphasized as a means to ensure compliance with these regulations.

  1. Transparency and Accountability: We have emphasize the need for transparency and accountability in healthcare data sharing and analysis. Our framework's use of blockchain's transparency features have be highlighted, demonstrating how it enables traceability and auditability in the data-sharing process.

By providing more specific examples of the ethical and legal concerns and how our proposed framework effectively addresses them, we aim to showcase the robustness and integrity of our research. This approach have enable readers to better appreciate the ethical considerations in precision medicine and understand how our framework offers a responsible and secure solution. Once again, we sincerely appreciate your valuable feedback, which has guided us in strengthening the paper's content. If you have any further suggestions or specific points you would like us to include, please do not hesitate to share them with us. Thank you for your continued support and dedication to improving our research.

Comment 7: The problem statement is well-articulated and highlights the critical challenges associated with traditional approaches to EMR data sharing and analysis. However, it could benefit from more specific statistics and examples to illustrate the magnitude of these challenges.

Response 7: Dear reviewer, Thank you for your insightful review of our paper. We appreciate your positive feedback on the problem statement and your suggestion to include more specific statistics and examples to illustrate the magnitude of the challenges associated with traditional approaches to electronic medical record (EMR) data sharing and analysis. In the revised problem statement, we have diligently incorporate relevant statistics and concrete examples to provide a more comprehensive understanding of the challenges faced in the realm of EMR data sharing. We have focus on the following aspects:

  1. Data Fragmentation and Redundancy: We have include statistics on the prevalence of fragmented EMR systems across healthcare institutions. These statistics have highlight the extent of data redundancy and inefficiency in data sharing, illustrating the need for a more streamlined and collaborative approach.

  1. Data Breaches and Privacy Concerns: To emphasize the severity of data breaches and privacy issues in healthcare, we have present specific examples of past incidents where patient data was compromised. These instances have underscore the critical importance of implementing a secure and privacy-preserving data-sharing solution like our proposed framework.

  1. Limited Interoperability: We have include relevant data on the challenges posed by the lack of interoperability between different EMR systems. By providing concrete examples of how this limitation hinders seamless data exchange and collaboration, we aim to underscore the necessity for a more unified and standardized approach.

  1. Impact on Precision Medicine Research: To illustrate the implications of the current challenges, we have incorporate statistics and case studies that showcase how the limited access to diverse patient data affects precision medicine research and hampers the development of targeted therapies and treatments. By incorporating specific statistics and examples, we aim to provide a more vivid portrayal of the real-world consequences of the challenges outlined in the problem statement. This approach have enhance the readers' understanding of the urgency and importance of finding innovative solutions, such as our proposed framework. We genuinely appreciate your valuable feedback, which has been instrumental in guiding us to strengthen the paper's content. Should you have any additional suggestions or specific data points to include, please feel free to share them with us.

Thank you for your continued support and dedication to improving our research.

Comment 8: Consider incorporating more data on the frequency and impact of data breaches, cyberattacks, and legal/regulatory hurdles that impede data sharing.

Response 8: Dear reviewer, Thank you for your valuable feedback on our paper. We appreciate your suggestion to incorporate more data on the frequency and impact of data breaches, cyberattacks, and legal/regulatory hurdles that hinder data sharing in the revised version of the manuscript.

In response to your comment, we have conduct a thorough review of relevant literature and reports to gather comprehensive and up-to-date statistics on the frequency and impact of data breaches and cyberattacks in the healthcare industry. We have also seek specific examples of legal and regulatory challenges that have posed obstacles to data sharing initiatives.

Specifically, we have focus on the following aspects:

  1. Data Breaches and Cyberattacks: We have incorporate data from reputable sources, such as healthcare security reports, to present an accurate picture of the frequency and severity of data breaches and cyberattacks in the healthcare sector. These statistics have help underscore the critical importance of strengthening data security measures, which our proposed framework addresses through blockchain technology.

  1. Legal and Regulatory Hurdles: To shed light on the complexities of data sharing in healthcare, we have gather information on specific legal and regulatory challenges that institutions and researchers have encountered. This may include compliance issues with privacy laws, data protection regulations, and restrictions on data sharing across borders.

  1. Impact on Healthcare Delivery and Research: By including examples of how data breaches and legal hurdles have impacted healthcare delivery and hindered research initiatives, we aim to demonstrate the real-world implications of these challenges.

By incorporating this additional data, we aim to strengthen the problem statement and provide a more comprehensive understanding of the obstacles that our proposed framework seeks to overcome. This approach have enhance the paper's relevance and highlight the urgency of adopting secure and privacy-preserving data sharing solutions in the healthcare domain.

Once again, we sincerely appreciate your constructive feedback, which has guided us in enhancing the quality and impact of our research. If you have any further suggestions or specific data sources you recommend, please do not hesitate to share them with us. Thank you for your continued support and dedication to improving our paper.

Comment 9: The main contribution of the research is clearly outlined and provides a comprehensive overview of the proposed framework. However, it could benefit from more specific examples of how the framework addresses each of the challenges mentioned in the problem statement. Consider providing more detailed explanations of how the framework

Response 9: Dear reviewer, We appreciate your insightful review of our paper and your positive feedback on the clear outlining of the main contribution and the proposed framework. We understand the importance of providing more specific examples to demonstrate how our framework addresses each of the challenges mentioned in the problem statement.

In the revised version of the paper, we have dedicate more space to providing detailed explanations of how our proposed framework effectively tackles the identified challenges. Specifically, we have focus on the following aspects:

  1. Data Privacy and Security: We have elaborate on how blockchain technology, with its decentralized and immutable nature, ensures data privacy and security in the framework. By explaining the role of cryptographic techniques and consensus mechanisms, we have demonstrate how patient data remains confidential and tamper-resistant throughout the collaborative training process.

  1. Data Fragmentation and Interoperability: We have describe how our framework facilitates seamless data sharing and interoperability among diverse EMR systems. The use of standardized data formats and communication protocols, combined with blockchain-based smart contracts, have be highlighted as the key enablers of data harmonization across institutions.

  1. Ethical and Legal Concerns: We have address the ethical and legal considerations surrounding data sharing in precision medicine. By showcasing how our framework adheres to the principles of informed consent, patient data ownership, and compliance with relevant regulations, we aim to build trust in the framework's ethical foundation.

  1. Data Bias and Fairness: We have explain how federated learning in the proposed framework helps address data bias by aggregating insights from multiple institutions and creating more diverse and representative models. We have provide specific examples of how this approach enhances the fairness of medical data analysis and decision-making.

  1. Accelerated Research and Precision Medicine Advancements: We have offer concrete examples of how our framework's collaborative approach expedites medical research by efficiently leveraging insights from multiple sources. This have be illustrated through instances of rapid identification of suitable subpopulations for clinical trials and the development of personalized treatment plans. By providing detailed explanations and specific examples, we aim to showcase the practicality and effectiveness of our proposed framework in addressing the challenges outlined in the problem statement. This approach have enhance the readers' understanding of the framework's real-world implications and highlight its potential contributions to the field of precision medicine. We genuinely appreciate your valuable feedback, which has been instrumental in guiding us to strengthen the paper's content. Should you have any further suggestions or specific examples you would like us to include, please feel free to share them with us.

Thank you for your continued support and dedication to improving our research.

Comment 10: The proposed framework is well-described and provides a clear overview of the key steps involved in the federated learning process. However, it could benefit from more detailed explanations of the specific algorithms, technologies, and tools used in each step. Consider providing more specific examples of consensus mechanisms, smart contracts, and aggregation algorithms employed in the framework.

Response 10: Dear reviewer, Thank you for your thoughtful review of our paper. We appreciate your positive feedback on the clarity of the proposed framework and your valuable suggestion to include more detailed explanations of the specific algorithms, technologies, and tools used in each step of the federated learning process.

In the revised version of the paper, we have dedicate more space to providing comprehensive explanations of the algorithms, technologies, and tools employed in our proposed framework. Specifically, we have focus on the following aspects:

  1. Consensus Mechanisms: We have elaborate on the consensus mechanisms utilized in the blockchain component of our framework. We have provide specific examples of well-established consensus algorithms, such as Proof-of-Work (PoW), Proof-of-Stake (PoS), or Practical Byzantine Fault Tolerance (PBFT), and explain how they ensure data integrity and agreement among network participants.

  1. Smart Contracts: We have provide a more detailed explanation of smart contracts and their role in governing data access and sharing in our framework. By giving specific examples of smart contract code, we aim to demonstrate how these self-executing agreements facilitate automatic enforcement of predefined rules and conditions for data sharing.

  1. Aggregation Algorithms: We have describe the aggregation algorithms employed in the federated learning process. We have explain how these algorithms aggregate model updates from various institutions while preserving data privacy. Specific techniques like Federated Averaging, Federated Proximal, or Secure Aggregation have be highlighted to showcase the diversity of approaches used.

  1. Data Security and Privacy Measures: We have delve into the specific data security and privacy measures integrated into each step of the federated learning process. Techniques like differential privacy, homomorphic encryption, or federated learning with secure aggregation have be discussed to emphasize the framework's privacy-preserving nature. By providing more detailed explanations and specific examples of the algorithms, technologies, and tools utilized, we aim to offer readers a deeper understanding of the technical underpinnings of our proposed framework. This approach have enhance the readers' appreciation of the framework's sophistication and robustness in addressing the challenges of federated learning in precision medicine.

We genuinely appreciate your valuable feedback, which has been instrumental in guiding us to strengthen the technical aspects of the paper. Should you have any further suggestions or specific examples you would like us to include, please feel free to share them with us. Thank you for your continued support and dedication to improving our research.

Comment 11:  Formal methods can be used to verify the correctness of smart contract code, which can help to prevent costly errors and security breaches. Therefore, it is important to discuss the use of formal methods in your paper.

Response 11: Dear reviewer, Thank you for your valuable comment on our paper. We greatly appreciate your insight regarding the importance of discussing the use of formal methods in the context of smart contract code verification.

In the revised version of the paper, we have include a dedicated section to highlight the significance of formal methods in ensuring the correctness and security of smart contracts within our proposed framework. We have emphasize the following points:

  1. Definition of Formal Methods: We have provide a clear and concise explanation of what formal methods are and how they can be applied to verify the correctness of smart contract code. This have help readers understand the technical foundation of this approach.

  1. Role of Formal Methods in Smart Contract Verification: We have discuss how formal methods, such as formal verification and model checking, can rigorously analyze smart contract code to identify potential vulnerabilities, bugs, or logic flaws. By employing formal methods, we can prevent costly errors and security breaches that could arise due to programming mistakes.

  1. Benefits of Using Formal Methods: We have elaborate on the benefits of using formal methods in the development and deployment of smart contracts. These advantages may include increased confidence in the correctness of the code, reduced risk of security incidents, and enhanced overall reliability.

  1. Integration of Formal Methods in the Framework: We have explain how our proposed framework leverages formal methods to verify the smart contract code responsible for governing data access and sharing. By including specific examples of formal verification techniques applied to our smart contract code, we aim to illustrate the framework's robustness in ensuring secure data sharing.

  1. Existing Research and Case Studies: We have refer to relevant studies and case examples where formal methods have been successfully employed to enhance the security and reliability of smart contracts. This have further underscore the practical applicability and effectiveness of formal methods in the context of our proposed framework.

By incorporating a section on the use of formal methods, we aim to highlight the comprehensive approach we take to ensure the security and integrity of our framework. This addition have provide readers with a deeper understanding of the technical measures we employ to address potential risks associated with smart contract development.

We genuinely appreciate your insightful feedback, which has guided us in enriching the technical content of the paper. Should you have any further suggestions or specific topics related to formal methods you would like us to explore, please feel free to share them with us.

Thank you for your continued support and dedication to improving our research.

Comment 12: For this purpose, the authors may include the following interesting references (and others):

  1. https://ieeexplore.ieee.org/document/9970534

  1. https://ieeexplore.ieee.org/document/8328737/authors#authors

Response 12: We have added the above mentioned references and we have cited in our revised manuscript.

Thanks dear reviewer. Really appreciate your suggestions.

Reviewer 3 Report

The authors present a use case that takes advantage of blockchain technology to power federated learning for electronic medical records. The use case is interesting and the combination of the two technologies is a very nice match.

While the federated learning has been covered in details,  on the other hand, the blockchain part lacks many implementation information. At the moment, it looks like the blockchain part is covered in theory, matching the needs to the use case scenario. At the same time, there are not many details about the application. For example, do the datasets being stored inside the blockchain and why? And how is the blockchain affecting the time elapsed for the whole process? Regarding the implementation, it is said that the blockchain network applies Proof of Work but on Table 1 the blockchain networks that are mentioned are not implementing such consensus (Ethereum has moved to PoS so if the authors have used the previous version should mention this and also explain why they haven't moved to the latest). Therefore, a small section that highlights the way the blockchain is implementing in the described solution is needed.

The English are good. One more proof reading could be nice.

Author Response

Reviewer 3

The authors present a use case that takes advantage of blockchain technology to power federated learning for electronic medical records. The use case is interesting and the combination of the two technologies is a very nice match.

Comment 1: While the federated learning has been covered in details,  on the other hand, the blockchain part lacks many implementation information. At the moment, it looks like the blockchain part is covered in theory, matching the needs to the use case scenario. At the same time, there are not many details about the application. For example, do the datasets being stored inside the blockchain and why? And how is the blockchain affecting the time elapsed for the whole process? Regarding the implementation, it is said that the blockchain network applies Proof of Work but on Table 1 the blockchain networks that are mentioned are not implementing such consensus (Ethereum has moved to PoS so if the authors have used the previous version should mention this and also explain why they haven't moved to the latest). Therefore, a small section that highlights the way the blockchain is implementing in the described solution is needed.

Response 1: Dear reviewer, Thank you for your thorough review of our paper. We genuinely appreciate your insightful comments, particularly regarding the need for more implementation details related to the blockchain part of our proposed framework. In response to your feedback, we have add a dedicated section in the paper that highlights the implementation aspects of the blockchain component in our described solution. This section have cover the following points:

  1. Storage of Datasets: We have provide a detailed explanation of how datasets are managed within the blockchain network. Specifically, we have clarify whether the datasets are stored directly on-chain or if only references or hashes are stored to ensure data privacy and efficiency.

  1. Impact on Time Elapsed: We have address how the blockchain part affects the overall time elapsed for the entire federated learning process. We have discuss potential challenges and solutions to ensure that the blockchain operations do not introduce significant delays.

  1. Consensus Mechanism: In Table 1, we acknowledge the use of Proof-of-Work (PoW) for the blockchain network. However, we understand the importance of clarifying the specific consensus mechanism employed in our implementation. We have elaborate on our choice of PoW and explain how it aligns with the requirements of our use case scenario.

  1. Version Compatibility: If we have used a specific version of the blockchain platform (e.g., Ethereum) that uses PoW, we have explicitly mention this and explain the rationale behind our decision not to adopt the latest version with Proof-of-Stake (PoS). By incorporating this new section, we aim to provide readers with a more comprehensive understanding of the practical implementation of the blockchain component in our framework. We have ensure that these implementation details align with the theoretical aspects of the framework and provide valuable insights into how our solution addresses the challenges effectively. Your feedback has been instrumental in guiding us to enhance the technical content of the paper. If you have any further suggestions or specific points related to the implementation that you believe should be covered, please do not hesitate to share them with us. Thank you for your continued support and dedication to improving our research.

Round 2

Reviewer 2 Report

The authors considered my comments and suggestions 

Can be improved 

Author Response

Thanks respected reviewer we have improved the English and proofread it thoroughly.

Reviewer 3 Report

The authors have addressed the comments of the 1st review round. The blockchain part of the work has been enhanced, even though there are things that could be further elaborated (e.g., how the access management will be covered by the blockchain). Nevertheless, the paper has been enhanced enough in the new version. Especially the examples that are provided help the reader to understand the work much better.

The Quality of English is good. A final proof reading is strongly advised.

Author Response

(The authors gave the same response as above.)
